# Macro-connectomics and microstructure predict dynamic plasticity patterns in the non-human primate brain

Sean Froudist-Walsh[1,2], Philip GF Browning[1,2,3], James J Young[4], Kathy L Murphy[5], Rogier B Mars[6,7], Lazar Fleysher[8], Paula L Croxson[1,2,9]*

[1]Department of Neuroscience, Icahn School of Medicine at Mount Sinai, New York, United States; [2]Friedman Brain Institute, Icahn School of Medicine at Mount Sinai, New York, United States; [3]Laboratory of Neuropsychology, National Institute of Mental Health, Bethesda, United States; [4]Department of Neurology, Friedman Brain Institute, Icahn School of Medicine at Mount Sinai, New York, United States; [5]Comparative Biology Centre, Medical School, Newcastle University, United Kingdom; [6]Centre for Functional MRI of the Brain, Nuffield Department of Clinical Neurosciences, John Radcliffe Hospital, University of Oxford, Oxford, United Kingdom; [7]Donders Institute for Brain, Cognition and Behaviour, Radboud University, Nijmegen, The Netherlands; [8]Department of Radiology, Icahn School of Medicine at Mount Sinai, New York, United States; [9]Department of Psychiatry, Icahn School of Medicine at Mount Sinai, New York, United States

*For correspondence:
paula.croxson@mssm.edu

Competing interests: The authors declare that no competing interests exist.

**Abstract** The brain displays a remarkable ability to adapt following injury by altering its connections through neural plasticity. Many of the biological mechanisms that underlie plasticity are known, but there is little knowledge as to when, or where in the brain plasticity will occur following injury. This knowledge could guide plasticity-promoting interventions and create a more accurate roadmap of the recovery process following injury. We causally investigated the time-course of plasticity after hippocampal lesions using multi-modal MRI in monkeys. We show that post-injury plasticity is highly dynamic, but also largely predictable on the basis of the functional connectivity of the lesioned region, gradients of cell densities across the cortex and the pre-lesion network structure of the brain. The ability to predict which brain areas will plastically adapt their functional connectivity following injury may allow us to decipher why some brain lesions lead to permanent loss of cognitive function, while others do not.
DOI: https://doi.org/10.7554/eLife.34354.001

## Introduction

Lesions to the brain set off a cascade of degenerative and protective plasticity-related processes. Distant grey matter degeneration, and a loss of anatomical connectivity of grey matter areas not directly affected by the lesion are common anatomical consequences of a lesion (*Catani and ffytche, 2005*; *Zaczek et al., 1980*). In addition, the extent of functional disconnection of intact regions is associated with the degree of behavioral impairment following a lesion (*Corbetta et al., 2005*; *He et al., 2007*) even if the areas remain structurally connected (*van Meer et al., 2010*). Conversely, some brain areas adapt by altering their connectivity patterns and increasing their connections with other, often unaffected areas (*Yogarajah et al., 2010*). It is thus important to be able to predict the areas that will undergo a relative functional disconnection following a lesion, and to predict which areas may functionally adapt in order to identify potential avenues for guiding adaptive plasticity.

**eLife digest** The brain has the ability to adapt after injury, a process known as plasticity. When one area sustains damage, for example following a car accident or stroke, other areas change their activity and structure to compensate. Understanding how this happens is critical to helping people recover from brain injuries. Certain factors may affect how well the brain can repair itself. These include how much the damaged area interacts with other areas, and which cell types different areas of the brain contain.

Froudist-Walsh et al. set out to determine how these factors influence recovery from brain injury in monkeys, whose brains are similar to our own. The monkeys had damage to a structure called the hippocampus. This part of the brain has a key role in memory, which is often impaired in patients with brain injuries. The hippocampus cannot repair itself because the brain has only a limited capacity to grow new neurons. Instead, the brain attempts to compensate for disruption to the hippocampus via changes in other, undamaged areas.

Using brain imaging, Froudist-Walsh et al. show that the types of changes that occur depend on how much time has passed since the injury. In the first three months, many areas of the brain change how much they coordinate their activity with other areas. Highly connected areas reduce their communication with other areas the most. In the long-term, the responses of brain areas depend more on which cell types they contain. Areas with more support cells known as "glia" – which supply nutrients and energy to neurons – are better able to adapt their connectivity up to a year after the injury.

These findings may ultimately benefit people who have suffered brain injuries after accidents or stroke. They suggest that stimulating intact brain areas may be helpful in the months immediately after an injury. By contrast, long-term therapy may need to focus more on structural repair. Future studies must build on these results to discover the best ways to induce successful recovery from brain injury.

DOI: https://doi.org/10.7554/eLife.34354.002

Currently, there is no quantitative way of predicting how unlesioned brain areas will adapt to injury elsewhere.

Recovery following brain injury is highly variable and occurs in stages. Much recovery occurs in the first few weeks following an injury, but functional improvements may continue until much later (*Berthier et al., 2011*; *Smania et al., 2010*). Studies in rodents have shown that the microstructural consequences of brain injury can vary dramatically at different times following injury, which could have serious implications for potential treatment strategies (*Hoskison et al., 2009*). However, small animal models of brain injury are not optimal for investigating chronic plastic changes, due to the short lifespan of rodents leading to a conflation of lesion- and neurodevelopmental- or aging-related plasticity. In human studies, pre-lesion scans are rare and are mostly available in patients with pre-existing brain abnormalities, such as patients with epilepsy. In studies of humans with brain lesions, the presence of possible pre-lesion pathology, combined with the non-specific nature of naturally occurring lesions, complicates interpretation. Consequently, little is known about how plasticity that occurs in the chronic stage following injury may differ from that occurring in the acute stage, and when particular functional and structural adaptations may take place.

The rapid advance in tools for measuring brain structure and function has lead to a great increase in the number of potentially informative predictors of plasticity following injury. It has recently been proposed that mapping a lesion onto an atlas of connections could predict the remote areas affected and perhaps the behavioral consequences of a lesion (*Kuceyeski et al., 2014*; *Thiebaut de Schotten et al., 2015*). While this approach could be greatly informative, it is not yet clear which remote areas may suffer the permanent negative consequences of an injury, and which may adapt and recover. Other studies have suggested that the role of brain regions within the whole brain architecture may be informative for the vulnerability to injury, with hub regions seemingly more likely to be affected in a variety of brain disorders (*Crossley et al., 2014*). This suggests the hypothesis that hub regions may distribute resources following a brain injury in order to aid recovery in areas

that are primarily affected by the injury (*Achard et al., 2012*). How this may occur at a microstructural level is unclear.

Recently, there has been a resurgence in interest in large-scale gradients in cortical organization (*Beul et al., 2017*; *Burt et al., 2018*; *Goulas et al., 2018*; *Margulies et al., 2016*; *Markov et al., 2014*; *Sanides, 1962*), and how this may enable cortical areas to specialize for distinct cognitive functions (*Chaudhuri et al., 2015*). However, little attention has been paid to whether cortical gradients of microstructural quantities, such as neuronal densities, or glial densities may also impose critical limits on the ability of an area to adapt to injury. Neuron densities vary smoothly across the cortical surface, with prefrontal cortex having less than half the neuron density of V1 (*Collins et al., 2010*). Non-neuronal cells such as astrocytes and microglia can have both beneficial and detrimental effects on post-injury plasticity (*Anderson et al., 2003*; *Loane and Kumar, 2016*), and the exact distribution of these cells throughout the brain may also constrain or modulate the response of a region to injury.

We set out to investigate whether it is possible to predict plastic changes following a discrete, specific lesion, using a bilateral excitotoxic lesion of the hippocampus. The hippocampus is a key part of the episodic memory circuit, but the impact of lesions restricted to the hippocampus itself is not always large (*Malkova and Mishkin, 2003*; *Zola-Morgan and Squire, 1986*). Because of the widespread nature of the episodic memory circuit (*Aggleton and Brown, 1999*), we hypothesized that this may be due to functional plasticity in the form of intact brain regions compensating for the damaged area (a process we previously showed to be critically dependent on cholinergic inputs to inferior temporal cortex following hippocampal disconnection (*Browning et al., 2010*; *Croxson et al., 2012*).

We acquired MRI scans in macaque monkeys before and at two time points after bilateral excitotoxic hippocampal lesions and found that the brain reacts to injury in a highly dynamic way, which is in part predictable on the basis of the pre-lesion functional connectivity and micro- and macro-structural anatomy. Areas that were most connected to the hippocampus before the lesion reduced their functional connectivity with areas in other modules in the acute stage, and showed a greater loss of grey matter volume during the chronic stage. Nonetheless, they increased their functional connectivity with other areas in the same module during the chronic stage, suggesting that highly dynamic processes of degeneration and plasticity occur in parallel over the year following the lesion. In contrast, hub regions suffered a general loss of functional connectivity during both the acute and chronic stages. Areas with a higher density of neurons lost connectivity with areas within the same module over the chronic period, while those with a higher density of non-neuronal cells (including glia and cardiovascular support cells) significantly increased their between-module functional connectivity over the same period, suggesting that a high density of these cells may be important to the plastic recovery process. This is the first study to demonstrate quantitatively a relationship between pre-lesion functional connectivity and the dynamic course of plasticity following a lesion and shows that information across a range of spatial scales can aid in prediction of the plastic recovery process following a lesion.

## Results

### Hippocampal lesions were precise and extensive

There was a significant reduction in hippocampal volume bilaterally during the acute stage measured by T2-weighted scans (*Figure 1A*), histologically (*Figure 1B*) and deformation-based morphometry of T1-weighted structural MRI scans (*Figure 1C–D*). All three analysis methods gave consistent results. Lesions were mostly bilateral and extensive, although there was some apparent sparing of the right hippocampus posteriorly across the five monkeys (*Figure 1*; *Table 1*). However, this is likely an under-estimation of the amount of damage in the posterior part of the hippocampus, which is narrower and therefore more susceptible to partial volume effects with neighbouring tissue.

### Functional and structural measures of plasticity

We measured structural and functional changes across the whole brain using high-resolution MRI at three time points: pre lesion, 3 months-post lesion and 12 months post-lesion in five macaque monkeys. The pre-lesion scans also included data from three additional control animals that did not go

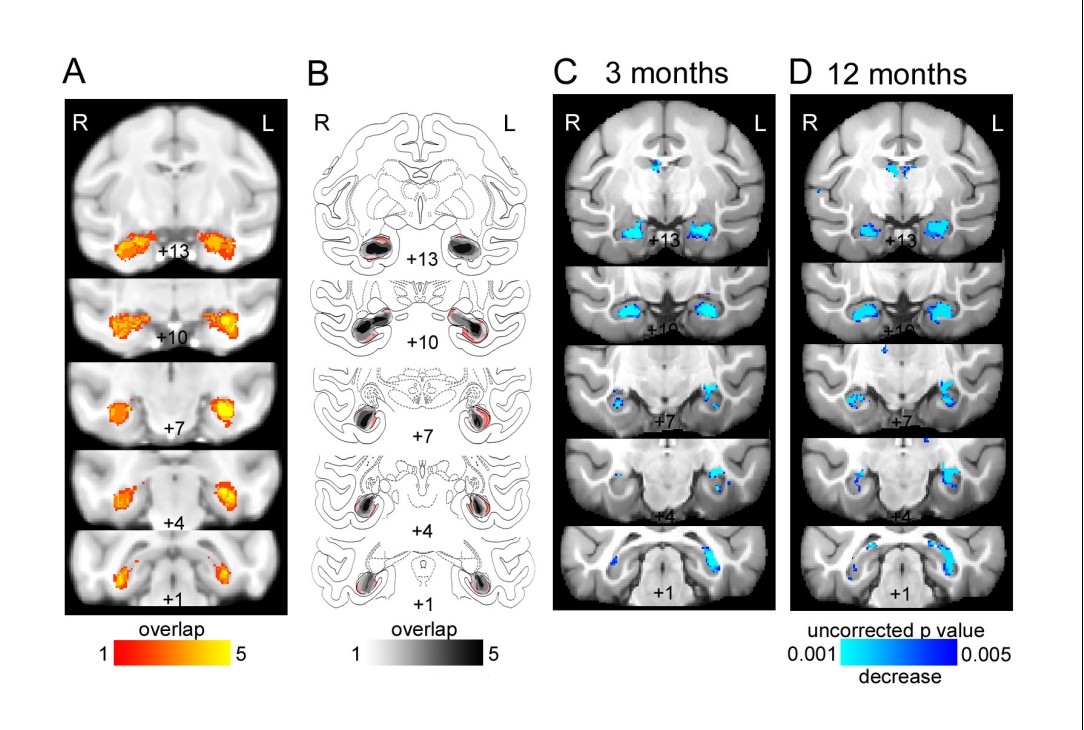

**Figure 1.** Bilateral hippocampal lesions. (**A**) T2-weighted hypersignal 6 days after surgery indicating local inflammation in the hippocampus; overlap is shown for the five monkeys. (**B**) Sketch of hippocampal size based on histology (Nissl stained sections) overlaid on atlas sections. The unlesioned hippocampal volume is shown in red. Overlap of remaining hippocampal volume is shown for the five monkeys indicating shrinkage of the hippocampus bilaterally in all monkeys. (**C-D**) Results of deformation-based morphometry analysis showing atrophy of the hipppocampus (**C**) 3 months after the lesion and (**D**) 12 months after the lesion.
DOI: https://doi.org/10.7554/eLife.34354.003

on to receive lesions. For clarity, we refer to the following stages: **acute** (pre-lesion vs. 3 months post-lesion) and **chronic** (3 months vs. 12 months post-lesion). We do not make any claims as to different rates of behavioral recovery during these stages, and acknowledge that cognitive recovery can occur in either stage following a brain insult (*Berthier et al., 2011*; *Lazar and Antoniello, 2008*).

Across all pairwise connections between brain regions, there was an overall increase in the functional connectivity strength over the acute stage ($t_{6318} = 9.37$, $p = 1 \times 10^{-22}$), and a decrease over the chronic stage ($t_{6318} = -16.85$, $p = 2 \times 10^{-62}$). In order to understand the specific regional

**Table 1.** Remaining volume of tissue in each lesioned monkey (calculated relative to atlas volumes from Nissl-stained histological sections registered to atlas sections) and lesion extent expressed as a percentage (1-(remaining volume/normal volume)).

| Monkey | Left hemisphere remaining volume | Right hemisphere remaining volume | Bilateral remaining volume | Left hemisphere lesion % | Right hemisphere lesion % | Total lesion % |
|--------|--------|--------|--------|--------|--------|--------|
| Atlas | 268144 | 268665 | 536809 | | | |
| Mean | 149379 | 121895 | 271274 | 44.29 | 54.63 | 49.46 |
| E | 184498 | 146047 | 330545 | 31.19 | 45.64 | 38.42 |
| M | 129741 | 84877 | 214618 | 51.62 | 68.41 | 60.02 |
| N | 159535 | 158787 | 318322 | 40.50 | 40.90 | 40.70 |
| S | 94769 | 89512 | 184281 | 64.66 | 66.68 | 65.67 |
| T | 178354 | 130254 | 308608 | 33.49 | 51.52 | 42.51 |

DOI: https://doi.org/10.7554/eLife.34354.010

changes that were driving these global effects, we first divided the brain into multiple 'modules', based on the resting-state functional connectivity data using the Louvain algorithm (*Blondel et al., 2008*). Here a 'module' is a set of brain regions that have higher functional connectivity with the other brain regions within the set than with brain regions outside the set. We investigated plastic changes to the mean within-module functional connectivity for each brain area, and to the network participation coefficient, which is a measure of how evenly the connections of a brain area are distributed across all of the modules in the brain. Thus brain regions that have a low proportion of their connections with brain regions outside the local module have a low network participation coefficient, whereas brain regions that are strongly connected with regions outside the local module have a high network participation coefficient. On this basis, the network participation coefficient has been proposed as a marker of connector hubs (*Power et al., 2013*). By analyzing within-module functional connectivity and the network participation coefficient, we can build a picture of the changes in processing within and between functional modules over time. As these methods depend on the definition of the modules, which in turn depends on a rather arbitrary choice of a resolution parameter (lambda), we report results that were robust across the entire range tested (minimum gamma = 0.8, corresponding to two brain modules, maximum gamma = 1.4, corresponding to just one brain region per module). Additionally, using deformation-based morphometry, we assessed changes to grey-matter volume over the acute and chronic stages.

## Anatomical and functional predictors of plasticity

We identified four factors that we hypothesized to be potential predictors of plasticity following the lesion.

First and second, at a cellular level, post-lesion plasticity depends on the ability of neurons to form novel synaptic connections, and on glial cells (particularly astrocytes and microglia) and other cardiovascular support cells to aid in the creation and maintenance of such synapses. We thus investigated whether the *gradients of neuronal and non-neuronal cell densities* across the cortex were associated with plasticity patterns in the acute and chronic stages. To do this, we mapped neuronal and non-neuronal cell densities from a macaque anatomical study (*Collins et al., 2010*) onto the Regional Map macroscopic template (*Kötter and Wanke, 2005*) (*Figure 2A,B*).

Third, studies in humans have suggested that hub regions are strongly affected following a range of neurological and psychiatric disorders, and that these regions are radically reorganized following injury (*Achard et al., 2012*; *Crossley et al., 2014*). We therefore investigated whether the hub-like properties of an area could predict its plastic alterations following hippocampal injury.

We created a continuous measure of the degree to which brain areas were hubs (a.k.a. '*hubness*') using the following method. As both network participation coefficient and node strength are proposed measures of hubness, and are positively correlated, we performed a principal components analysis on the strength and participation coefficient data, and took the first principal component, which explained 73.19% of the variance in strength and participation coefficient to be our estimate of hubness (*Figure 2C*).

Fourth, we reasoned that the strength of *pre-lesion functional connectivity with the hippocampus* (the lesioned region) should affect the degree to which other regions in the brain plastically reorganize their functional connectivity following the lesion, with regions that were highly functionally connected with the hippocampus likely being most highly affected by the lesion, and consequently most in need of plastic reorganization.

We assessed pre-lesion hippocampal functional connectivity with all other cortical regions on the basis of the pre-lesion resting-state fMRI scans and averaged between left and right hippocampus. The average hippocampal functional connectivity is shown in *Figure 2D*. The strongest functional connectivity was with medial and ventral temporal regions that are in close proximity to the hippocampus. In contrast, dorsal frontal regions showed a slight negative correlation with the hippocampus. In order to test the anatomical validity of these functional connectivity patterns, we compared them to the anatomical connectivity measures from the original (*Stephan et al., 2001*) and 'enhanced' (*Deco et al., 2014*) versions of the CoCoMac tract-tracing atlas. The enhanced version has previously been a better fit to functional connectivity measures than the discrete-valued version of the tract-tracing atlas (*Deco et al., 2014*; *Grayson et al., 2016*). Hippocampal functional connectivity measured in the current study was highly correlated with anatomical connectivity measures in

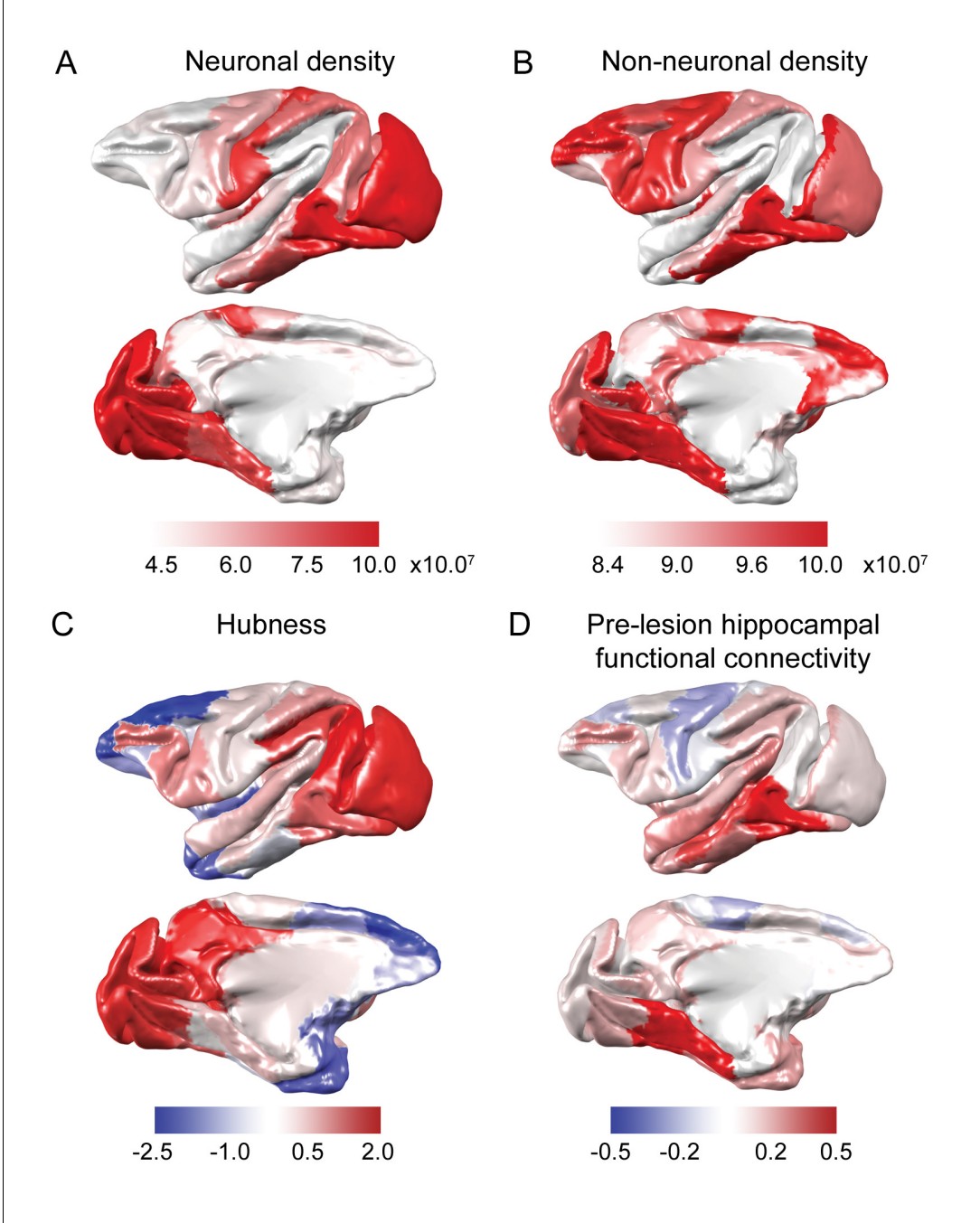

**Figure 2.** Anatomical and functional predictors of plasticity. (A-B) Neuron and non-neuronal cell densities were mapped from *Collins et al., 2010*. (C) Hubness was calculated as each area's projection onto the first principal component of node strength and network participation coefficient data. (D) Pre-lesion hippocampal functional connectivity was strongly correlated with anatomical connectivity derived from the CoCoMac tract-tracing atlas (r = 0.54, p = $2.2 \times 10^{-7}$) and enhanced versions of the CoCoMac Atlas (r = 0.60, p = $3.6 \times 10^{-9}$). The hippocampus was most strongly connected to ventral temporal lobe structures..

DOI: https://doi.org/10.7554/eLife.34354.004

both the original (r = 0.54, p = $2.2 \times 10^{-7}$) and enhanced versions of the CoCoMac Atlas (r = 0.60, p = $3.6 \times 10^{-9}$).

## Hubness, and pre-lesion hippocampal functional connectivity predict an acute stage drop in network participation

We entered the neuronal density, non-neuronal cell density, hubness and pre-lesion hippocampal functional connectivity as predictors of acute changes in network participation coefficient in a stepwise regression (*Figure 3A*). The model significantly predicted the cortex-wide pattern of acute changes in network participation coefficient, explaining over half of the variance ($F_{2,75} = 42.24$, $p = 5 \times 10^{-13}$, $r^2 = 0.53$, *Figure 3B*). Hubness ($t_{73} = -5.70$, $p = 2 \times 10^{-7}$), and pre-lesion hippocampal functional connectivity ($t_{73} = -5.25$, $p = 1 \times 10^{-6}$) were significantly associated with a drop in network participation coefficient over the acute stage (*Figure 3C*). Neither neuron density ($t_{73} = -1.04$, $p = 0.29$) nor non-neuronal cell density showed a significant association ($t_{73} = -0.55$, $p = 0.59$).

## Non-neuronal cell density predicts a chronic stage rise in network participation

As both the calculation of the acute and chronic stage changes contained the three-month timepoint, they were not independent. In order to identify the chronic stage changes that were independent of the acute stage changes, we constructed a general linear model, using the acute stage changes to predict the chronic stage changes. The relationship between the acute and chronic stage changes to the network participation coefficient did not differ from chance ($p = 0.66$, corrected for the shared timepoint, see Materials and methods), suggesting that distinct degenerative and plastic processes affected network participation at the two stages. The residuals of this model were taken to be the chronic stage changes that were independent of the acute stage changes.

We used the same predictors as during the acute stage to predict the chronic stage changes in the network participation coefficient (*Figure 3D*). The model significantly predicted the cortex-wide pattern of chronic stage changes in network participation coefficient ($F_{2,75} = 24.3$, $p = 7 \times 10^{-9}$, $r^2 = 0.39$, *Figure 3E*). Non-neuronal cell density was significantly associated with a rise in network participation coefficient over the chronic stage ($t_{73} = 4.70$, $p = 1 \times 10^{-5}$). As in the acute phase, hubness was significantly associated with a drop in network participation coefficient during the chronic stage ($t_{73} = -4.93$, $p - 4 \times 10^{-6}$) (*Figure 3F*). Neither neuron density ($t_{73} = 1.57$, $p = 0.12$) nor hippocampal functional connectivity ($t_{73} = 1.24$, $p = 0.22$) were significant predictors of chronic stage network participation changes.

## Hubness is associated with an acute stage drop in within-module functional connectivity

A model with hubness as the lone predictor (identified with stepwise regression) significantly predicted the cortex-wide pattern of acute stage changes in within-module functional connectivity (*Figure 4B*; *Figure 4C*, $F_{1,76} = 10.25$, $p = 0.002$, $r^2 = 0.12$, hubness $t_{73} = -3.20$, $p = 0.002$, neuron density $t_{73} = -1.39$, $p = 0.168$, non-neuronal cell density $t_{73} = -0.92$, $p = 0.359$, pre-lesion hippocampal functional connectivity $t_{73} = -0.28$, $p = 0.782$).

## Higher pre-lesion hippocampal functional connectivity is associated with a chronic stage rise in within-module functional connectivity

Acute (*Figure 4A*) and chronic stage (*Figure 4D*) changes to within-module connectivity were more strongly positively associated than expected by chance ($p < 0.001$, corrected for the shared timepoint, see Materials and methods), suggesting that there may have been a continuation of degenerative or plastic processes from the acute to chronic stage. The residuals of this model were used to identify the independent chronic stage changes to within-module connectivity.

The stepwise regression model significantly predicted the cortex-wide pattern of chronic stage changes in within-module functional connectivity ($F_{3,74} = 7.68$, $p = 0.0002$, $r^2 = 0.24$, *Figure 4E*). Neuron density ($t_{73} = -2.54$, $p = 0.013$) was a significant predictor of a drop in within-module functional connectivity during the chronic stage. In contrast, pre-lesion hippocampal functional connectivity was associated with a chronic stage increase in within-module functional connectivity ($t_{73} = 4.04$, $p = 0.0001$). Non-neuronal cell density was not included in the final model ($t_{73} = -1.33$, $p = 0.19$). Hubness ($t_{73} = -2.08$, $p = 0.04$) was a significant predictor of a chronic stage decrease in within-module functional connectivity across most (lambda = 0.8–1 and 1.3–1.4), but not all

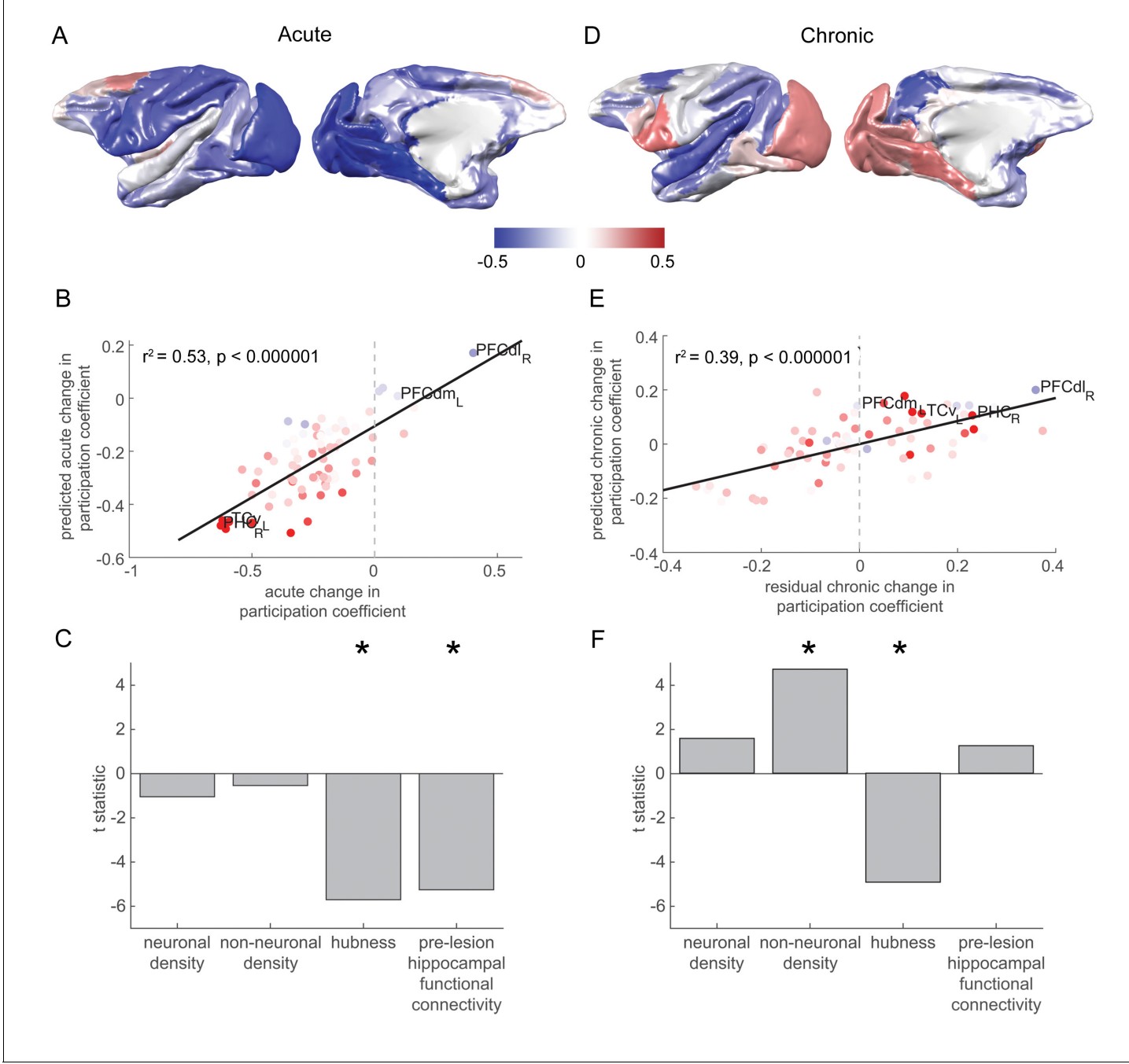

**Figure 3.** Changes in network participation are strongly predicted by pre-lesion anatomy and functional connectivity. (A) Most brain regions showed a drop in network participation over the acute stage. (B-C) The degree to which individual brain regions reduced their network participation over the acute stage was well predicted by their pre-lesion connectivity to the hippocampus and the extent to which they acted as hubs in the pre-lesion network ('hubness'). (B) A scatter plot of the acute stage changes to the network participation coefficient for each brain regions, compared to model predictions. Brain regions are coloured according to their pre-lesion connectivity with the hippocampus (compare with *Figure 2D*). (D-F) As in (A-C), but for chronic stage changes to network participation. Note that areas with a higher non-neuronal cell density showed the greatest increase in the network participation coefficient over the chronic stage. Note that (D) shows the overall within network participation coefficient changes for the chronic stage, while the model predictions and data shown in (E-F) corresponded to the residual chronic stage changes to within the network participation coefficient, after regressing out acute stage changes. * signifies that these predictors were significant and included in the final model.
DOI: https://doi.org/10.7554/eLife.34354.005

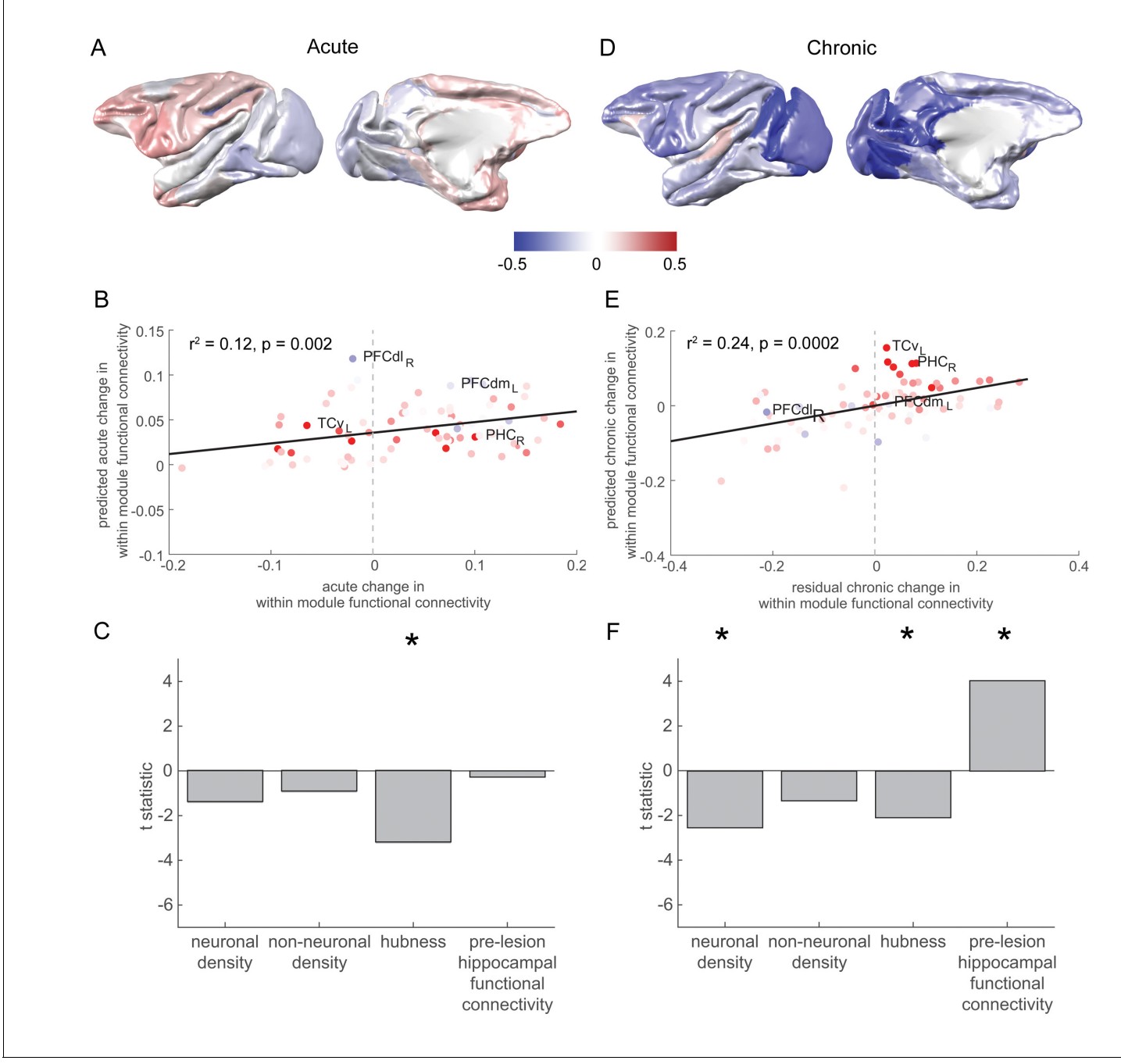

**Figure 4.** Pre-lesion hippocampal connectivity is associated with a rise in within-module connectivity over the chronic stage. (A) The pattern of acute stage increases and decreases to within-module connectivity. (B-C) The degree to which individual brain regions changed their within-module connectivity over the acute stage was significantly associated with the extent to which they acted as hubs in the pre-lesion network ('hubness'). Scatter plot in (B) shows brain regions coloured according to their pre-lesion connectivity with the hippocampus (compare with *Figure 2D*). (D-F) As in (A-C), but for chronic stage changes to within-module connectivity. Note that (D) shows the overall within module-connectivity changes for the chronic stage, while the model predictions and data shown in (E-F) corresponded to the residual chronic stage changes to within module-connectivity, after regressing out acute stage changes. * signifies that these predictors were significant and included in the final model.

DOI: https://doi.org/10.7554/eLife.34354.006

(lambda = 1.1, p = 0.053, lambda = 1.2, p = 0.075) of the repetitions of the analysis with different resolution parameters (lambda), and thus may be viewed as a marginal result (*Figure 4F*).

## Higher pre-lesion hippocampal functional connectivity is associated with a chronic stage drop in grey-matter volume

None of the four predictors significantly predicted acute stage changes in grey matter volume (neuron density: $t_{73} = 1.10$, $p = 0.27$, non-neuronal cell density: $t_{73} = 0.40$, $p = 0.69$, hubness: $t_{73} = 1.27$, $p = 0.21$, hippocampal functional connectivity: $t_{73} = -0.11$, $p = 0.91$) (*Figure 5A–C*).

Acute and chronic stage changes to cortical grey matter volume were more strongly positively associated than expected by chance ($p = 0.01$, corrected for the shared timepoint, see Materials and methods), suggesting that there may have been a continuation of degenerative grey matter loss from the acute to chronic stage. The residuals of this model were used to identify the independent chronic stage changes to cortical grey matter volume.

The stepwise regression model for chronic stage changes to grey matter volume included only pre-lesion hippocampal functional connectivity in the final model ($F_{1,76} = 16.39$, $r^2 = 0.17$, $t_{73} = -4.05$, $p = 0.0001$). Neuron density ($t_{73} = 1.58$, $p = 0.118$), non-neuronal cell density ($t_{73} = 0.28$, $p = 0.779$) and hubness ($t_{73} = -1.92$, $p = 0.058$) did not make the cut-off for inclusion in the model (*Figure 5E–G*).

## Grey matter volume was reduced in a small number of areas in the acute stage, with further reductions in the chronic stage

In order to investigate grey-matter volume changes to the whole-brain (not restricted to cortex), we performed a deformation-based morphometry analysis of the grey-matter volume changes using a linear mixed model. Results are shown in *Figure 5D,H*, thresholded at $p < 0.005$ and a minimum cluster size of 5 mm$^3$ (*Sallet et al., 2011*).

During the acute stage, there were very limited volumetric decreases in the medial septum, amygdala and dorsal premotor cortex. No increases survived thresholding (*Figure 5D*). At the chronic stage, we still saw decreases in the medial septum, but also a larger range of decreases. Some of these were also in areas that are monosynaptically connected with the hippocampus: the medial orbitofrontal cortex, posterior cingulate cortex and posterior parahippocampal cortex. There were also more extensive volumetric decreases, in the anterior prefrontal cortex (medial and lateral), ventrolateral prefrontal cortex, dorsal striatum, visual cortex and superior temporal cortex. There were also volumetric increases in the cerebellum, midbrain and premotor cortex. These results did not survive multiple comparisons correction (possibly due to our small sample size), so these changes should be viewed with this caveat in mind.

## Modularity is affected by hippocampal lesions

We investigated the effect that the hippocampal lesions had on the macroconnectivity structure by examining the changes in individual modules, which, along with hubs are considered the canonical forms of integration and segregation, and hallmarks of interareal connectomes (*Rubinov, 2016*).

We estimated modules based on the pre-lesion data, repeated 10,000 times. The most reliable modules are shown in *Figure 6A*. Four modules were identified, orbitofrontal cortex/anterior temporal lobe, posterior temporal, parieto-occipital and dorsal frontal. The relative functional connectivity of these modules within the pre-lesion network is shown in *Figure 6B* (colors as in 6A). In these force-directed graph representations, functional connectivity acts as an attractive force between two nodes, so nodes that are closer together are more highly connected. The parieto-occipital module (orange) is highly connected to the three other modules.

Following the orange parieto-occipital module through time (*Figure 6B*), three months after the lesion the parieto-occipital module is still relatively closely connected both with itself and with the other three networks (although its nodes have dispersed a little). At 12 months after the lesion however, this module (orange) becomes dramatically dispersed.

We quantified this dispersion as the mean drop in within-module functional connectivity over the acute and chronic stages (*Figure 6C–D*). During the acute stage, (*Figure 6C*), there is not a large drop in within-module functional connectivity, although there is a significant difference between modules ($F_{3,76} = 19.54$, $p = 4 \times 10^{-9}$), with the parieto-occipital module dispersing somewhat, and the

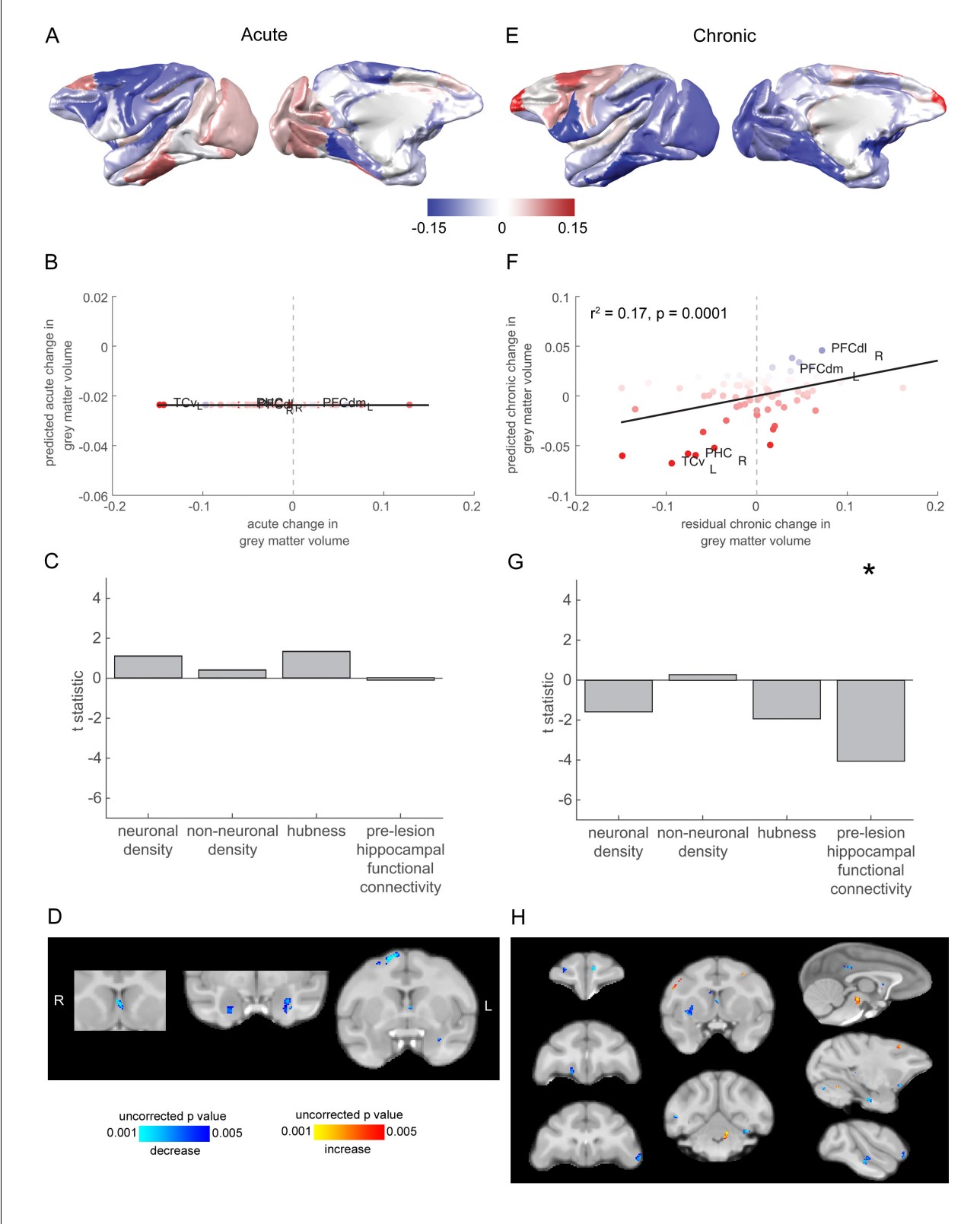

**Figure 5.** Grey matter loss at the chronic stage was most prominent in cortical regions that were strongly connected to the hippocampus. (A) The pattern of acute stage increases and decreases to cortical grey matter volume. (B-C) No significant predictors of acute stage cortical grey matter changes were identified. (D) Whole brain voxelwise analysis revealed very limited volumetric decreases in the medial septum, amygdala and dorsal premotor cortex. (E) As in (A), but for chronic stage changes to cortical grey-matter volume. (E-F). The degree to which individual cortical regions

*Figure 5 continued on next page*

*Figure 5 continued*

changed their grey-matter volume over the chronic stage was significantly associated with the extent to which they were functionally connected to the hippocampus before the lesion. Scatter plot in (B) shows brain regions coloured according to their pre-lesion connectivity with the hippocampus (compare with *Figure 2D*). Note that (E) shows the overall within module-connectivity changes for the chronic stage, while the model predictions and data shown in (F-G) corresponded to the residual chronic stage changes to within module-connectivity, after regressing out acute stage changes. * signifies that this predictor was significant and included in the final model. (H) Whole brain voxelwise analysis revealed a larger range of decreases in grey-matter volume over the chronic stage. There were also volumetric increases in the cerebellum, midbrain and premotor cortex. These results did not survive multiple comparisons correction.

DOI: https://doi.org/10.7554/eLife.34354.007

dorsal frontal and OFC/anterior temporal modules increasing their within-module functional connectivity.

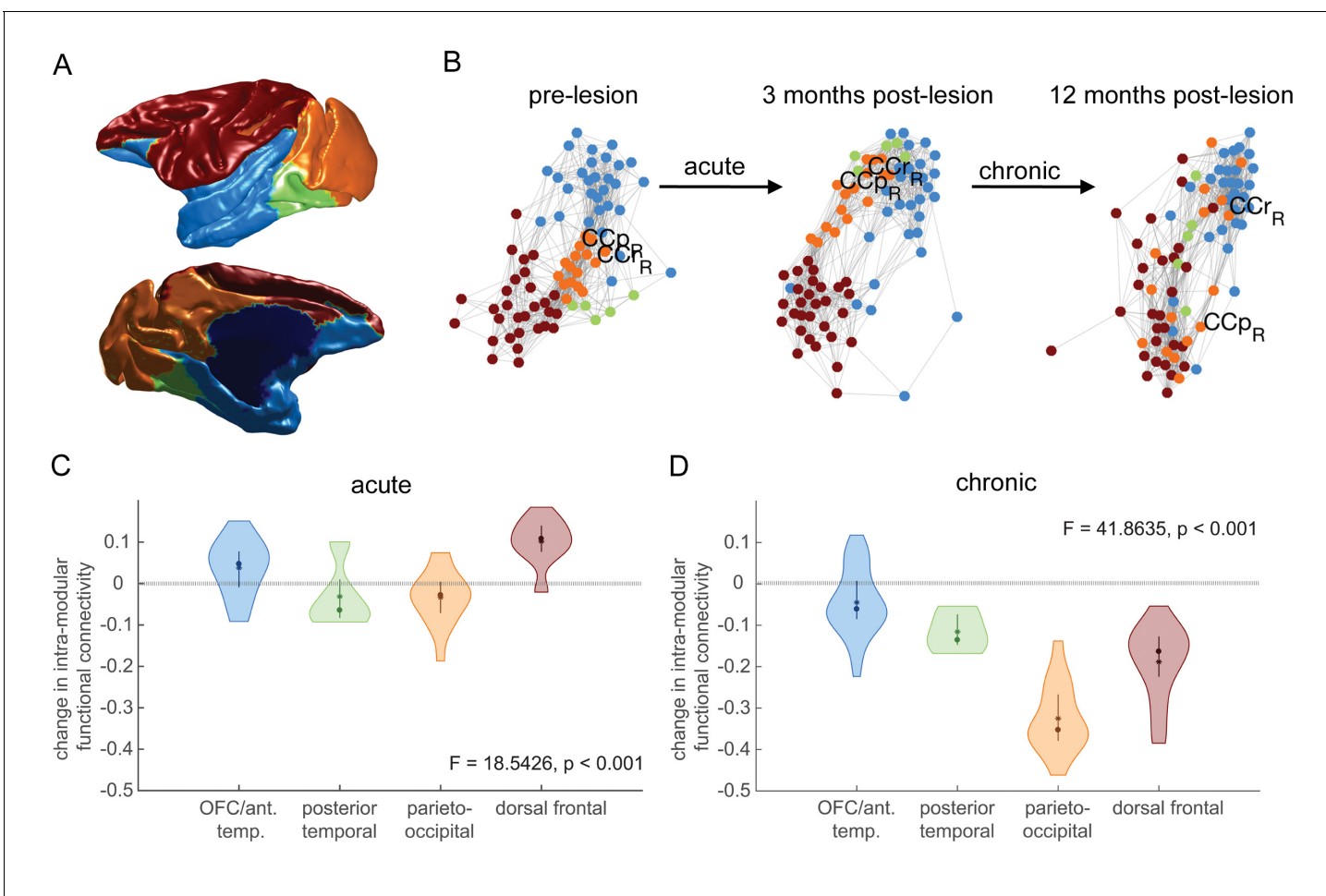

**Figure 6.** Effects of hippocampal lesions on module structure. The most consistent pre-operatively defined modules are shown anatomically (A) and using a force-directed graph representation (B) where highly functionally connected brain regions are plotted close together. Before the lesion, the parieto-occipital module (orange) is highly connected to the three other modules. At 3 months post-lesion, the network looks largely similar, although there may have been some dispersion. At 12 months post-lesion however, the parieto-occipital module is completely dispersed, with some dispersion of other modules. (C-D) We quantified this dispersion by the mean change in within-module functional connectivity for each module. There is a significant effect of module on node dispersion over both the acute (C – pre-lesion vs. 3 months post-lesion) and chronic (D – 3 months vs 12 months post-lesion) stages.

DOI: https://doi.org/10.7554/eLife.34354.008

During the chronic stage (*Figure 6D*), there is a drop in within-module functional connectivity across all modules, with by far the greatest dispersion occurring in the parieto-occipital module ($F_{3,76}$=41.86, p = $4\times10^{-16}$).

The dispersion of the parieto-occipital module, which previously acted as a link between other modules, also led to a drop in connectivity between the modules, as seen by an increase in the modularity during the chronic stage (pre-lesion: 0.33, 3 months post-lesion: 0.34, 12 months post-lesion: 0.49; $F_{2,297}$=60230, p~0). There were always fewer modules at 12 months after the lesion compared to the pre-lesion scan, at all values of lambda tested (range 0.8–1.4).

### Relationship between cell densities and hubs

In an exploratory analysis we performed a stepwise regression in order to assess the relationship between cell densities and 'hubness'. A model containing both neuron and non-neuronal cell density significantly predicted hubness ($F_{2,75}$ = 6.18, $r^2$ = 0.14, p = 0.003). Neuron density was positively associated with hubness (t = 3.49, p = 0.008), while non-neuronal cell density was negatively associated with hubness (t = −2.19, p = 0.032).

### Statistical comparison between full dataset and two-monkey dataset

In order to test whether the above results were due to the fact that, in some cases, scans of different monkeys were used at different timepoints, we repeated all analyses with the data from two monkeys with a complete set of pre- and post-lesion scans. We tested whether the beta values of the four independent variables (neuron and non-neuronal cell densities, hubness and pre-lesion hippocampal connectivity) for each of the above regression analyses significantly differed between the two datasets using non-parametric statistics (see Materials and methods). Of the 24 beta-values assessed, the following differences in beta-values between datasets were observed. Hippocampal connectivity as a predictor for chronic changes in participation coefficient (p = 0.034), hubness as a predictor of acute changes in within-module connectivity (p = 0.0004) and hubness as a predictor of chronic changes in grey matter volume (p = 0.019). In all three of these cases the significance of the result, and hence the interpretation did not change; that is, hubness remained a significant predictor of acute changes in within-module connectivity in the full and two-monkey datasets, and the two other predictors remained non-significant.

## Discussion

Here we show that, following an excitotoxic lesion of the hippocampus, functional and structural changes occur dynamically over time. The pattern of functional connectivity of the hippocampus before a lesion can significantly predict the brain areas that will show functional connectivity changes and grey matter volume loss after the lesion. Areas that were highly connected with the hippocampus before the lesion showed a drop in network participation during the acute stage, and a loss of grey matter volume over the chronic stage. However, they also increased their functional connectivity with other areas in the same module over the chronic stage. This may indicate that MRI is sensitive to distinct microscale plasticity processes that occur during the acute and chronic stages post-lesion. This was supported by contrasting associations between measures of neuronal and non-neuronal cell densities with functional connectivity changes. Neuronal density was associated with a greater loss of within-module functional connectivity during the chronic stage, while non-neuronal cell density was associated with increases in network participation over the same stage. Network hubs showed a distinctive pattern of post-lesion alterations, which suggested they are more vulnerable to the effects of the lesion. The more a region acted as a hub in the pre-lesion network, the greater the reduction in functional connectivity at both the acute and chronic stages.

### Pre-lesion functional connectivity predicts dynamic network participation alterations

Changes in activation in distant brain regions are widely agreed to be among the first adaptations following brain injury before a return to more normal-appearing activation patterns in the spared tissue close to the affected site (*Cramer, 2008*). Recently in humans, connectomic information derived from databases of healthy humans has been used to identify remote areas likely to be affected by a lesion (*Kuceyeski et al., 2014*; *Thiebaut de Schotten et al., 2015*), but the relationship between

the brain's connectivity profile and the dynamics of plasticity had not previously been investigated. Here, we show that functional connectivity can be highly predictive of dynamic plastic changes in both the acute and chronic stages.

Stronger pre-lesion functional connectivity of the hippocampus was associated with a *drop* in network participation over the acute stage and cortical grey matter volume over the chronic stage. The strength of connectivity with the hippocampus was also associated with an *increase* in within-module connectivity over the chronic stage. An intriguing possibility that should be investigated in larger future studies is that the functional connectivity loss and recovery may correspond to the timeline of loss and recovery of behavioral function following a lesion.

With just three scans over a year-long period, we were able to detect interesting dynamics of local and global plasticity. This begs the question: at which stage over the year following injury are the majority of changes happening? In a recent study, Grayson and colleagues examined network changes in functional connectivity during reversible chemogenetic suppression of amygdala activity (*Grayson et al., 2016*). In the minutes and hours following amygdala suppression, they were already able to detect some network changes, with strong reduction in functional connectivity of the amygdala and its local modules. On visual comparison of the force-directed graph plots of the current study (*Figure 6*) and the study of Grayson et al (their Figure 7), it appears that the global network structure is better preserved following chemogenetic disruption of the amygdala, than following permanent lesions of the hippocampus. This is not particularly surprising as permanent lesions are more likely to induce large-scale plasticity, on the timescales of the present study. In that study they did not explicitly calculate changes in hub functional connectivity or other graph-theory properties presented here, limiting the ability to directly compare results. Nonetheless, future studies combining chemogenetic inactivations and permanent lesions of the same brain regions with multiple scanning timepoints have the potential to uncover the fine-grained timeline of global brain alterations following disruption of brain regions, and disentangle how these two methods for interfering with the function of an area may induce different alterations to brain functional connectivity and plasticity.

Our findings also have implications for the understanding of the role of medial temporal lobe structures in memory function. While lesions of the hippocampus have been implicated in human amnesia for decades (*Corkin et al., 1997*; *Scoville and Milner, 1957*), it is also clear that even focal hippocampal damage has widespread consequences beyond the immediate functional damage, and that memory is a distributed process that contains both segregation and overlap of function (*Gaffan, 2002*). This concept of connectional diaschisis (*Carrera and Tononi, 2014*) has also been identified in human patients with focal hippocampal damage, where the functional alterations to a network extended far beyond the structural damage (*Henson et al., 2016*). We significantly extended these findings by quantitatively predicting changes in whole-brain functional connectivity and grey matter volume from pre-lesion hippocampal functional connectivity, microstructural gradients and network-based brain measures.

The processes underlying the mechanism of an excitotoxic lesion are partly overlapping with those involved in human brain injury. The initial phase of an ischemic event, for example, leads to excitotoxic death via activation of glutamate receptors, as in our deliberate NMDA lesion, but this is only one of a cascade of processes (*Cramer, 2008*). The mechanisms involved in traumatic brain injury are less similar to our excitotoxic lesions, starting with cerebral edema and increased intracranial pressure, followed by a number of other factors of which glutamate excitotoxicity is just one (*Kinoshita, 2016*). The strength of a specific NMDA-induced lesion, which spares fibers of passage within or adjacent to the area (*Coffey et al., 1988*; *Köhler and Schwarcz, 1983*) is that we can study the effect of damage to a specific area on the rest of the brain.

## Grey matter volume loss is in connected areas of the episodic memory network

None of our factors predicted grey matter loss over the acute stage, but areas that were highly connected with the hippocampus before the lesion suffered a greater loss of grey matter volume over the chronic stage. We observed decreases in the volume of the medial septum, amygdala and posterior parahippocampal cortex. These regions and the white matter tracts connecting them to the hippocampus are also affected in human subjects with developmental amnesia (*Dzieciol et al., 2017*; *Olsen et al., 2013*) and in people born very preterm (*Ball et al., 2012*; *Caldinelli et al., 2017*; *Froudist-Walsh et al., 2017*; *Salvan et al., 2014*; *Tseng et al., 2017*). Although the severe structural

abnormalities associated with developmental amnesia lead to seemingly permanent impairments to episodic memory (*Vargha-Khadem et al., 2001*), milder damage to this circuit may enable plastic changes in cortical functional connectivity to partially compensate for damage to the core episodic memory circuit (*Isaacs et al., 2003*; *Nosarti and Froudist-Walsh, 2016*). In the present study, the incomplete damage to subcortical structures such as the mammillary bodies, fornix and connected thalamic subregions in combination with plastic changes to spared areas may be crucial for the preservation or recovery or anterograde memory abilities (*Baxter, 2013*; *Froudist-Walsh et al., 2018*; *Mitchell et al., 2008*). Nonetheless, we acknowledge a limitation of the study is that our design cannot distinguish between compensatory and maladaptive plasticity.

## Neuronal and non-neuronal cell densities predict different aspects of post-lesion plasticity

We did not see a significant relationship between neuronal or non-neuronal density and acute post-lesion plasticity. Neuronal cell density was significantly associated with the decrease in within-module functional connectivity during the chronic stage. Dendritic tree size and spine count tend to show opposite gradients to neuron density, with lowest values in early visual cortex and peaking in higher association areas (*Elston et al., 2010*; *Scholtens et al., 2014*). Thus the loss of connectivity in areas with higher neuron density may be reflective of other factors, such as a lack of dendritic spines that can be crucial for synaptic plasticity. Indeed, local and distant remodeling of spines and dendritic trees has been observed following stroke (*Brown et al., 2007*; *Brown et al., 2010*; *Nudo, 2013*).

We found that non-neuronal cell density was significantly positively associated with the increase in network participation during the chronic stage. Although synaptic plasticity is traditionally thought of as being neuronally initiated, it is now clear that astrocytes and microglia can modify synaptic connectivity in a variety of ways (*Ben Achour and Pascual, 2010*; *Allen and Barres, 2005*; *Araque et al., 1999*; *Ullian et al., 2004*) and can even alter synaptic strength in the absence of neuronal activity (*Clark et al., 2015*). Astrocytes and microglia can have both beneficial and detrimental effects on post-injury plasticity (*Anderson et al., 2003*; *Loane and Kumar, 2016*) and have emerged as promising candidates for treatment following acquired brain injury in humans (*Barreto et al., 2011*; *Loane and Kumar, 2016*). Our finding that non-neuronal cell density positively correlates with the increase in the network participation coefficient during the chronic stage provides a novel link between the local role of glia at the synapse, and plasticity of large-scale functional connectivity patterns.

## Hubs are preferentially affected by hippocampal lesions

We found that hub regions were more likely to lose functional connectivity with other regions (reflected in a drop in both within-module functional connectivity and network participation) following a lesion. This supports the idea that hubs are generally affected following brain injury or disorder. *Crossley et al. (2014)* put forward two hypotheses as to why hub regions are more likely to suffer pathology in brain disorders (*Crossley et al., 2014*). The first hypothesis stated that hub regions are more functionally valuable, and therefore damage to hub regions is more likely to be symptomatic than damage elsewhere. Here we show that hub regions are in fact more likely to suffer a loss of functional connectivity, even if the primary site of injury – the hippocampus – is not itself a hub. This coincides to a greater degree with the second hypothesis of Crossley et al., namely that hubs are biologically costly, and thus more vulnerable to various pathogenic processes. An extension of that hypothesis is that hubs are more likely to be connected to the site of primary insult (in this case the hippocampus), and more likely to suffer from diaschisis as a result.

We showed that 'hubness' was an independent predictor of structural and functional losses following a lesion, even after accounting for the effects of functional connectivity to the lesioned area. The degradation of the hubs was also associated with a destruction of the overall network structure in the chronic stage. At 12 months following the lesion, the whole brain network had separated into a smaller number of weakly interconnected modules. This demonstrates effects that focal lesions can have on global brain function.

The mechanism underlying the vulnerability of hubs to injury requires further study. Speculatively, in an exploratory analysis, we found a relatively high neuron: non-neuronal cell ratio in hubs, perhaps indicating a lack of glial other support cells per neuron. This may mean that hubs are less able to

adapt to injury than non-hub areas. Several studies have recently examined the relationships between cell densities and hub properties yet consistent relationships have yet to emerge, perhaps due to the use of different experimental techniques and definitions of hubs across different species (*Beul et al., 2015*; *Beul et al., 2017*; *van den Heuvel et al., 2015*; *Rubinov et al., 2015*; *Scholtens et al., 2014*).

### Caveats and future directions

Given our finding that plasticity following a lesion is highly dependent on the cellular composition of different brain regions, it may be the case that the specific cellular composition of the hippocampus may also have played a role in the patterns of plasticity we see here. The unique connectivity of the hippocampus relative to other brain regions may also be an important factor. This suggests that we may see very different patterns of plasticity following lesions to other brain regions. Future studies will be needed to determine if the relationships between connectivity patterns, gradients of micro-structure and patterns of plasticity following brain injury can be generalised to lesions to other brain regions or are specific to the hippocampus.

### Conclusions

By combining precise anatomical lesions with multiple, multi-modal scans across a period of a year following a lesion, we were able to make three contributions to the literature. Firstly, we show that functional and structural changes can greatly differ between acute and chronic stages. This highlights the importance of carefully considering the time since injury when studying post-lesion plasticity and behavioral recovery. We advocate, where possible, the collection of data at multiple time points following injury in order to accurately map the dynamic recovery process. Secondly, while it has been known for some time that areas connected to a lesioned brain region are more likely to be affected by the lesion than non-connected areas, to our knowledge this is the first study to show quantitatively that post-lesion plasticity patterns depend on pre-lesion functional connectivity. Lastly, we link across spatial scales, and show how microstructural gradients and macrostructural network measures can provide additional predictive value and insights into the plasticity process.

## Materials and methods

This study was conducted ethically, humanely, and in accordance with federal regulations and the guidelines contained in the National Research Council Guide for the Care and Use of Laboratory Animals. All animals were handled according to a protocol approved by the Animal Care and Use Committee (IACUC) at the Icahn School of Medicine at Mount Sinai. The institution also has been fully accredited by the Association for Assessment and Accreditation of Laboratory Animal Care International (AAALAC - #00002) since 1967.

### Data

Data are available to download from the INDI PRIMatE Data Exchange (*Milham et al., 2018*): https://www.nitrc.org/account/login.php?return_to=http://fcon_1000.projects.nitrc.org/indi/PRIME-downloads.html: Mount Sinai Philips Achieva 3T dataset. Users will first be prompted to log on to NITRC and will need to register with the 1000 Functional Connectomes Project website on NITRC (http://fcon_1000.projects.nitrc.org/indi/PRIME/mssm1.html) to gain access to the PRIME-DE datasets.

The code used for analysis has been made available on Github: https://github.com/seanfw/froudist-walsh-et-al-elife-2018 (*Froudist-Walsh, 2018*; copy archived at https://github.com/elifesciences-publications/froudist-walsh-et-al-elife-2018).

### Subjects

Subjects were seven male rhesus macaque monkeys (*Macaca mulatta*; mean age at start of experiment 3.5 years, range 2.9–4 years, mean weight at start of scanning 6.0 kg, range 4.7–7.2 kg), and one female cynomolgus macaque monkey (*Macaca fascicularis*; 8 years at start of experiment, 4.7 kg at start of scanning). 4 of the male monkeys and the female monkey received bilateral neurotoxic hippocampal lesions as described below. The animals were young adults at the time of lesion (mean

age for the males, 4.4 years, range 3.7–4.75 years; female age 8 years). The other three males acted as unoperated controls, along with pre-lesion data acquired before the lesions in the other monkeys. They were scanned at the same point in the behavioral study as the operated males. Full datasets were not available for every monkey as, due to the difficulty associated with acquiring high-resolution data from monkeys, some datasets were not of sufficient quality. The data acquired for each monkey is shown in *Table 2*.

## Behavior

All monkeys except the female cynomolgus were tested on a test of episodic memory, the object-in-place scene learning task. The behavioral results from these monkeys M, N, S and T are described elsewhere (H1-H4 in *Froudist-Walsh et al., 2018*). The monkeys had a retrograde memory impairment but no anterograde memory impairment on the episodic memory task.

## Hippocampal lesions

Monkeys received MRI-guided bilateral neurotoxic hippocampal lesions using methods described by Hampton et al (*Hampton et al., 2004*). Neurosurgical procedures were performed in a dedicated operating theatre under aseptic conditions. Briefly, monkeys were sedated with a cocktail of dex-medetomidine (0.01 mg/kg), buprenorphine (0.01 mg/kg) and midazolam (0.1 mg/kg) given i.m.. Where necessary, top-ups were given of dex-medetomidine (0.003 mg/kg) and midazolam (0.1 mg/kg) without buprenorphine (to avoid excessive respiratory depression) and any further top-ups of dex-medetomidine (0.003 mg/kg) only as necessary. This protocol was selected to avoid the use of the NMDA antagonist ketamine, which would potentially counteract the effects of the NMDA used as an excitotoxin (*Hampton et al., 2004*).

Monkeys were intubated, an i.v. catheter placed and anesthesia was maintained with sevoflurane (1.5–4%, to effect, in 100% oxygen). Monkeys were given glycopyrrolate (0.01 mg/kg i.m.), antibiotics (Cefazolin, 25 mg/kg i.m.), steroids (methylprednisolone, 20 mg/kg i.v.), non-steroidal anti-inflammatories (meloxicam, 0.2 mg/kg i.v.), and a H2 receptor antagonist (ranitidine, 1 mg/kg, i.v.) to prevent against gastric ulceration following the administration of both steroids and non-steroidal anti-inflammatories. Atipamezole was used to reverse the α2-adrenergic agonist if necessary, once anesthesia was stabilized. Monkeys received i.v. fluids throughout the procedure (5 ml/kg/hr i.v.).

The monkey was placed in a stereotaxic frame in exactly the same position as for the pre-operative structural MRI scan (employing a tooth marker; *Saunders et al., 1990*). The head was cleaned with antimicrobial cleaner and the skin and underlying galea were opened in layers. Small holes were drilled over the injection entry points: one dorsal and posterior to the long axis of the hippocampus and one dorsal to the uncus in each hemisphere (see *Hampton et al. (2004)* for details). Two micro-manipulators (Kopf Instruments, Tujunga, CA) were fitted with gas-tight syringes (Hamilton, Reno, NV) with a 28 ga needle, point style 4, using measurements obtained from the preoperative T1-weighted scan at the most anterior extent of the hippocampus and injections of N-methyl

**Table 2.** Details of monkeys and surgeries.

| Monkey | Group | T1-weighted | | | Resting-state | | |
|---|---|---|---|---|---|---|---|
| | | Pre | 3 month | 1 year | Pre | 3 month | 1 year |
| E | Lesion | X | X | | X | X | |
| M | Lesion | X | X | X | X | X | X |
| N | Lesion | X | X | X | | | X |
| S | Lesion | X | X | X | | X | X |
| T | Lesion | X | X | X | X | X | X |
| C | Control | X | | | X | | |
| L | Control | X | | | X | | |
| W | Control | X | | | X | | |
| Total | | 8 | 5 | 4 | 6 | 4 | 4 |

DOI: https://doi.org/10.7554/eLife.34354.009

D-aspartate (NMDA; 0.3 M in sterile saline) were made from anterior to posterior, spaced 1.5 mm apart. Each injection was 3 µl in volume, made at a rate of 0.5 µl/min, with 1 min between targets. After the final injection the needle was raised 0.5 mm and 10 min elapsed before it was extracted. For the uncus injections two injections per hemisphere were made, 3 µl in volume, made at a rate of 0.5 µl/min, with 3 min between targets. Propanolol (0.5 ml of 1 mg/ml per dose) was administered immediately prior to the NMDA injections and re-administered as necessary (up to four times) to prevent tachycardia during the injections due to nonspecific effects of NMDA. One monkey received propofol (4.0 ml total in boluses of 0.5–1.0 ml of a 10 mg/ml solution) to supplement anesthesia, due to tachypnoea, also likely to be a nonspecific effect of NMDA. Once the lesion was completed the skin and galea were sewn in layers.

When the lesion was complete, monkeys received 0.2 mg/kg metoclopramide (i.m.) to prevent postoperative vomiting. Monkeys also received 0.1 mg/kg midazolam (i.m.) to prevent seizures. They were extubated when a swallowing reflex was evident, returned to the home cage, and monitored continuously until normal posture was regained. Post-operatively monkeys were treated with antibiotics, steroids and analgesia for 3–5 days. Operated monkeys were returned to their social groups within 3 days of the surgery.

Following the first surgery we assessed the lesion extent with a T2-weighted scan (*Málková et al., 2001*) and used the result to plan the second surgery, targeting the injection coordinates to regions with low hypersignal. All monkeys received two lesion surgeries except monkey E, which only required one.

Whole-brain BOLD functional MRI data were collected for 40 min using a three-dimensional sequence with the following parameters: 40 axial slices; dimensions 1.5 × 1.5×1.5 mm; TR, 2600 ms; TE, 19 ms; 988 volumes, acceleration factor = 2. A structural scan (three averages) was acquired for each monkey using a T1-weighted magnetization-prepared rapid-acquisition gradient echo sequence (0.5 × 0.5×0.5 mm). An additional T1-weighted scan and a T2-weighted scan (0.5 × 0.5×0.5 mm) were acquired 6 days post-operatively to assess lesion extent. For the resting-state fMRI scans, isoflurane levels were kept to a minimum to ensure the preservation of resting-state networks: mean isoflurane 1.2%, range 1.0–1.6% (*Hutchison et al., 2014*; *Vincent et al., 2007*). Resting-state fMRI was carried out at least 2 hr after ketamine administration, to reduce detrimental effects of ketamine on resting-state networks (*Bonhomme et al., 2016*). End-tidal $CO_2$ was maintained in a normocapnic range wherever possible, to avoid effects of hypercapnia on the BOLD signal: mean CO2 39 mmHg, range 33–45 mmHg (*Bandettini and Wong, 1997*; *Kastrup et al., 1999*; *Rostrup et al., 2000*).

## Histology

At the end of the study, monkeys were deeply anaesthetized with ketamine (10 mg/kg), intubated and given sodium barbiturate (sodium pentobarbital, 100 mg/kg) intravenously. They were then transcardially perfused with 0.9% saline followed by 4% parafomaaldehyde. Brains were post-fixed in paraformaldehyde overnight and then cryoprotected in 30% sucrose solution in 0.9% saline and cut into 50 µm sections coronally on a freezing microtome. 1 in five sections was stained with cresyl violet for cell bodies. The sections containing the hippocampus were photographed using a Nikon Eclipse 80i light microscope with a 4x objective. Hippocampal volumetric reduction was carried out in Fiji, a version of the image analysis program ImageJ (https://imagej.nih.gov/ij/). The volume of the hippocampus was manually delineated on sections of the monkey atlas 'Red' (using criteria from *Málková et al., 2001*) and the remaining hippocampal volume of the hippocampus was manually delineated on images of the cresyl violet sections. The sections were then nonlinearly warped to the atlas using the function bUnwarpJ and the volume of each hippocampal section calculated as a percentage of normal hippocampal volume (*Table 1*). The overlap between the remaining hippocampal volume across all five monkeys and normal hippocampal volume is shown in *Figure 1B*.

## MRI data analysis
### Hippocampal lesion assessment and structural alterations in connected areas
#### Structural data preprocessing
This analysis involved structural scans from the pre-operative (n = 8), 3 months (n = 5) and 12 months (n = 4) time points. The three structural images were averaged together to produce an image with high signal to noise ratio.

#### Hippocampal lesion assessment
The post-operative T2-weighted scan was linearly registered to the pre-operative T1-weighted scan using the FSL tool FLIRT. The pre-operative T1-weighted scans were then nonlinearly registered to monkey MNI space using the FSL tool FNIRT and the inverse of the transform was also calculated. We manually defined a hippocampus region of interest (ROI) on the MNI brain according to criteria described in *Málková et al. (2001)*, registered it back to the individual monkey's pre-operative T1-weighted image and manually edited it for precision. After thresholding the T2-weighted image for each monkey at 85% of its maximum intensity and combining the two images (for the first and second surgery) for monkeys M, N, S and T, we calculated the overlap with the hippocampus ROI for each monkey. The volume of T2 hypersignal relative to the hippocampus ROI volume is shown in *Table 3*. The T2 overlap for all monkeys is shown in in *Figure 1A*.

## Deformation-based morphometric analysis
The structural data were first analyzed using a VBM-style analysis as employed by Sallet et al. (*Sallet et al., 2011*), using the tools FNIRT and Randomise (*Jenkinson et al., 2012*; *Winkler et al., 2014*). First, all brains were warped onto the MNI rhesus macaque atlas template (*Frey et al., 2011*) using the affine linear registration tool FLIRT and then the nonlinear registration tool FNIRT to produce a study-specific template image. Because the amount of warping expected from the pre-operative to 3 months and 12 months time points was disproportionately large due to the lesions, we included all off the brains, not just the control data, in the template (*Reuter and Fischl, 2011*; *Reuter et al., 2012*). The nonlinear warping underwent five iterations, each with a higher resolution warp and increasing refinement of the template, with the final warp using a warp resolution (knot-spacing of cubic b-splines) of 1 mm isotropic. The restricted log determinant of the Jacobian of the warp field for each brain to the template was extracted. This is the scalar value of the amount of directional stretching required to align each structural image with the template.

Voxelwise analysis was carried out on an area limited by a grey matter mask extracted using automated segmentation with FAST on the rhesus macaque MNI template (*Frey et al., 2011*). Longitudinal changes in grey matter volume were assessed using a linear mixed-effects model, implemented in Matlab with the FreeSurfer function lme_mass_fit_vw for mass-univariate linear mixed model analysis (*Bernal-Rusiel et al., 2013*) (https://surfer.nmr.mgh.harvard.edu/fswiki/LinearMixedEffectsModels). For consistency, only time points and monkeys for which we had resting-state data were analysed. Regions were designated as significant if they passed a threshold of $p < 0.005$, with a cluster extent threshold of 5 mm³ voxels (*Sallet et al., 2011*).

**Table 3.** Lesion volumes calculated from T2-weighted hypersignal relative to whole hippocampal volume for each monkey. All monkeys received two lesion attempts except monkey E.

| Monkey | Lesion attempts | Hippocampus | | | Left | | | Right | | |
|---|---|---|---|---|---|---|---|---|---|---|
| | | Volume | Lesion | % | Volume | Lesion | % | Volume | Lesion | % |
| E | 1 | 821.38 | 244.13 | 29.72 | 429.25 | 94.88 | 22.10 | 392.13 | 149.25 | 38.06 |
| M | 2 | 1019.75 | 563.38 | 55.25 | 516.63 | 190.50 | 36.87 | 503.13 | 372.88 | 74.11 |
| N | 2 | 1161.38 | 179.50 | 15.46 | 607.63 | 109.38 | 18.00 | 553.75 | 70.13 | 12.66 |
| S | 2 | 979.24 | 706.49 | 72.15 | 484.12 | 398.37 | 82.29 | 495.12 | 308.12 | 62.23 |
| T | 2 | 937.38 | 690.63 | 73.68 | 442.88 | 364.00 | 82.19 | 494.50 | 326.63 | 66.05 |

DOI: https://doi.org/10.7554/eLife.34354.011

## Network changes following hippocampal lesions

### Functional data preprocessing

Resting-state fMRI data were analyzed using tools from FMRIB Software Library (FSL) (*Jenkinson et al., 2012*), and MATLAB (Mathworks, Natick, MA, USA). Each functional dataset was first skull-stripped using BET (*Smith, 2002*) and manually corrected to make sure that all brain areas were included. Head motion artifacts were removed by linear regression (MCFLIRT), Gaussian spatial smoothing was applied (FWHM 3 mm) and high-pass filtering (0.01 Hz) was applied to reduce noise from scanner drift. Artifacts from vasculature, respiration and head motion were identified using independent components analysis restricted to 40 components (MELODIC) (*Beckmann and Smith, 2004*) and removed by linear regression.

### Tract-tracing data and atlas

Structural connectivity data were derived from the CoCoMac database of tract-tracing experiments, originally described by Stephan et al (*Stephan et al., 2001*), as well as an 'enhanced' version of the atlas, which was recently developed by Deco and colleagues (*Deco et al., 2014*). This enhanced version of the atlas was developed by iteratively re-weighting connection strengths, and resulted in dramatically improved fits to resting-state functional MRI data. The data were used in combination with the Regional Map atlas (*Kötter and Wanke, 2005*) which was developed by the same group who originally developed CoCoMac, for use with the database, and was mapped to the F99 standard monkey template by Shen and colleagues (*Shen et al., 2012*). The Regional map contains 82 mainly cortical regions, and includes the hippocampus and amygdala. The brain regions used, along with their abbreviations, are shown in *Table 4*.

A warp was calculated from the F99 brain to the monkeys MNI brain using ANTs normalization software (*He et al., 2007*) and was applied to the Regional Map atlas using nearest-neighbor interpolation in order to get the atlas into the (monkey MNI) space in which the functional MRI data resided.

### Connectivity matrix construction

Timecourses of hippocampal activity on the pre-lesion scans were extracted. Functional connectivity of the left and right hippocampus with all other regions from the pre-lesion scans was calculated and data for left and right hippocampus were averaged. The pre-lesion hippocampal connectivity was then used as a predictor of plastic changes between timepoints. The hippocampus and amygdala bilaterally were excluded from the network calculations (unless stated otherwise) in order to maintain a constant number of nodes in the network and allow comparison with cortical cell densities from the study of *Collins et al. (2010)*. Timecourses from each of the remaining atlas regions were then extracted from each resting-state scan. For each scan, a $78 \times 78$ functional connectivity matrix was constructed. All analysis, except for those looking at the force-directed graph representations, were carried out on the continuous, unthresholded functional connectivity matrices, in order to avoid choosing arbitrary thresholds (Karolis et al., 2016). Functional connectivity matrices were averaged across all monkeys at each timepoint.

### Cortical grey matter preparation

We then attempted to predict structural alterations throughout the brain during the acute and chronic stages using the same predictors as for the functional connectivity analyses described above. In order to do this, we used the warps from each original pre- and post-lesion scan to the template. Specifically, we extracted the log-Jacobian value for each area in the Regional Map, for each warp. The log-Jacobian is a value that represents the amount of contraction (for negative numbers) or expansion (positive numbers) that has taken place in order for the original brain region to closely match the same brain region on the atlas. If a region has reduced in volume during the acute stage, then this will be represented as a negative value when subtracting the log-Jacobian of the region on the pre-lesion scan from the equivalent value at the 3-month scan. In contrast, an increase in volume would lead to a positive value. The same logic holds from the chronic stage, when comparing the 3 month and 12 month scans.

**Table 4.** List of regional map abbreviations and corresponding brain areas.

| Regional map abbreviation | Brain area |
| --- | --- |
| A1 | Primary auditory cortex |
| A2 | Secondary auditory cortex |
| Ia | Anterior insula |
| Ip | Posterior insula |
| Amyg | Amygdala |
| CCa | Anterior cingulate cortex |
| CCp | Posterior cingulate cortex |
| CCr | Retrosplenial cortex |
| CCs | Subgenual cingulate cortex |
| FEF | Frontal eye field |
| G | Gustatory area |
| HC | Hippocampus |
| M1 | Primary motor cortex |
| PFCcl | Centrolateral prefrontal cortex |
| PFCdl | Dorsolateral prefrontal cortex |
| PFCdm | Dorsomedial prefrontal cortex |
| PFCm | Medial prefrontal cortex |
| PFCoi | Intermediate orbital prefrontal cortex |
| PFCol | Orbitolateral prefrontal cortex |
| PFCom | Orbitomedial prefrontal cortex |
| PFCvl | Ventrolateral prefrontal cortex |
| PFCpol | Polar prefrontal cortex |
| PHC | Parahippocampal cortex |
| PMCdl | Dorsolateral premotor cortex |
| PMCm | Medial (supplementary) premotor cortex |
| PMCvl | Ventrolateral premotor cortex |
| S1 | Primary somatosensory cortex |
| S2 | Secondary somatosensory cortex |
| PCi | Inferior parietal cortex |
| PCip | Cortex of the intraparietal sulcus |
| PCm | Medial parietal cortex |
| PCs | Superior parietal cortex |
| TCc | Central temporal cortex |
| TCi | Inferior temporal cortex |
| TCs | Superior temporal cortex |
| TCpol | Polar temporal cortex |
| TCv | Ventral temporal cortex |
| V1 | Primary visual cortex |
| V2 | Secondary visual cortex |
| VACd | Dorsal anterior visual cortex |
| VACv | Ventral anterior visual cortex |

DOI: https://doi.org/10.7554/eLife.34354.012

## Visualisation of hippocampal functional connectivity and comparison with CoCoMac

Hippocampal functional connectivity was visualized on the surface using the Connectome Workbench (www.humanconnectome.org/software/connectome-workbench).

After extracting hippocampal functional connectivity with the rest of the brain, we compared this to the hippocampal structural connectivity on the basis of the original and enhanced CoCoMac atlases, using Pearson correlations.

## Cell density analysis

In order to map neuronal and glial density data onto the atlas used in the current study, two authors (SFW and PLC) performed a consensus mapping from of the areas described in the anatomical study of Collins et al. (*Collins et al., 2010*) onto the Regional Map (*Kötter and Wanke, 2005*). If multiple areas from the Collins Map related to a single area in the Regional Map, then a simple average of all relevant areas was taken to represent the neuronal (or glial) density in that area.

## Functional connectivity measures

In order to parsimoniously assess how each brain region's functional connectivity profile was affected by the hippocampal lesion, we assessed two measures of functional connectivity. Within-module functional connectivity is the average functional connectivity of a node with all other nodes within its local module (a.k.a. module). We also analyzed changes to the network participation coefficient, which assesses how strongly a node's functional connections are distributed across all modules in the brain, and has been proposed as a measure of connector hubs in the brain (*Power et al., 2013*).

In order to assess brain functional connectivity in this manner, we defined the brain's modules (modules) using the Louvain module detection algorithm (*Blondel et al., 2008*), as implemented in the Brain Connectivity Toolbox (*Rubinov and Sporns, 2010*). Module detection was repeated 10000 times, and the most consistent modules were used in the rest of the manuscript.

Module detection using the Louvain algorithm depends on choice of a resolution parameter (lambda). In general results presented in the main manuscript use the default parameter value (1) unless otherwise stated. However, we repeated all analyses using a range of values (between 0.8 and 1.4). These values were chosen as they represented extremes of module detection, with just two modules being detected at lambda = 0.8, and modules of single brain areas being detected at lambda = 1.4.

## Hubs and modularity

As both high node strength and high network participation coefficient are proposed as identifiers of network hubs, and the two metrics were positively correlated in the present data, we performed a principal components analysis on the $78 \times 2$ matrix that represented a node strength and network participation coefficient for each grey matter region. We found that the first principal component explained 73.19% of the variance in the node strength and network participation coefficient data. This first principal component then became our continuous 'hubness' measure.

Using the previous definition of modules, we then assessed the longitudinal changes in within-module functional connectivity. A drop in within-module functional connectivity can be seen on the force-directed graph representation of the matrix as a dispersion of the modules over time (*Figure 6*). Differences in dispersion between modules in the acute and chronic stages were assessed using one-way ANOVA.

We then repeated the same analysis, but instead of assessing drop in within-module functional connectivity for each module, we compared hub to non-hub regions, again using a one-way ANOVA.

## Prediction of plasticity in acute and chronic stages

The relationship between neuron density, non-neuronal cell density, hubness, pre-lesion hippocampal connectivity and our outcome measures of plasticity (within-module functional connectivity, network participation and cortical grey matter volume) was assessed using a series of stepwise regression models.

For the acute stage changes, a simple stepwise regression was performed with neuron density, non-neuronal cell density, hubness, pre-lesion hippocampal connectivity as predictors and acute stage changes to either within-module functional connectivity, network participation and cortical grey matter volume as the dependent variable.

As the definition of the acute and chronic stage changes shared the 3 month scan, they were not independent. In order to identify the chronic stage changes that were independent of acute stage changes, we first assessed the relationship between the two using a general linear model with acute stage changes as the independent variable and chronic stage changes as the dependent variable. The residuals from this analysis were taken to be the independent chronic stage changes.

For the chronic stage changes, a stepwise regression was then performed, again with neuron density, non-neuronal cell density, hubness, pre-lesion hippocampal connectivity as predictors and the residual chronic stage changes to either within-module functional connectivity, network participation and cortical grey matter volume as the dependent variable.

As, by definition, the acute (3 month – 0 months) and chronic (1 year – 3 month) stages share a common timepoint (the 3 month scan), one would expect a negative correlation across regions by design. In order to assess whether the relationship between acute and chronic stage changes differed from that expected by chance, we created a null-distribution as follows. On each of 1000 simulations, 234 random numbers were drawn and randomly divided into three groups of 78, matching the number of extra-hippocampal brain regions recorded at the pre-lesion, acute and chronic stage scans, respectively. Randomly generated acute (3 month – 0 month) and chronic (12 month – 3 month) stage 'change' measures were then calculated for each simulation, and a general linear model was constructed with the random acute stage measure as a predictor of the random chronic stage changes across the 78 'regions'. For each of the 1000 simulations, a t-statistic for the random acute stage predictor was saved. These t-statistics became our null distribution, to which our experimentally observed t-statistic could be compared. The experimental relationship was considered significant if the value was within the top or bottom 2.5% of simulated correlation coefficients.

## Relationship between micro- and macro-scale connectivity measures

We performed an exploratory analysis, where we analysed the relationship between cell density measures and hubness. We used neuronal and non-neuronal cell densities as predictors of hubness in a stepwise regression.

## Statistical comparison between full dataset and two-monkey dataset

We tested for differences between the beta values in the regressions for the full and two-monkey datasets use non-parametric statistics as follows. Under the null hypothesis, there is no difference between the full and two-monkey datasets. Therefore we should obtain the same beta values for our regressions at the acute and chronic stages regardless of whether the dependent variable data (e.g. acute change in participation coefficient) for each region is taken from the full, or two-monkey dataset. We thus created a null distribution as follows. For each of the 78 brain regions, the acute change in participation coefficient was randomly taken from either the full, or two monkey datasets. We then ran the acute stepwise regression, as in the original analysis. The beta values for the four independent variables (neuron density, non-neuronal cell density, hubness and hippocampal connectivity) for the original (full data) regression were then subtracted from the beta values for regression with the randomly drawn data, and this difference in beta values was saved. This process was repeated 10,000 times with different random draws. Finally, to obtain a two-tailed p-value, the difference between the beta values for the regression on the two-monkey data from the beta-values for the full-data regression for each independent variable were compared to the null distribution. If the difference between the full- and two-monkey datasets lay outside the middle 95% of the null distribution, then the beta-values were said to be different at the $p < 0.05$ (two-tailed) level.

This process was repeated for each of the six dependent variables reported in the paper (acute/chronic stage changes in participation coefficient, within-module connectivity and grey matter volume).

## Acknowledgements

We would like to thank Mark Baxter, Hauke Kolster, Ronald Primm, Ignacio Medel, Frank Macaluso, Charles Adapoe, Cheuk Tang and Zahi Fayad for their support of this research. We would also like to thank Mark Baxter and Peter Rudebeck for their comments on an earlier version of this manuscript and Vyacheslav Karolis for helpful discussions. We would also like to thank Gustavo Deco and Kelly Shen for providing data relating to the CoCoMac database and the Regional Map atlas.

This work was supported by a Charles H Revson Foundation Senior Fellowship in the Biomedical Sciences to PLC and the Friedman Brain Institute at the Icahn School of Medicine at Mount Sinai. RBM is supported by a BBSRC David Phillips Fellowship (BB/N019814/1).

## Additional information

### Funding

| Funder | Author |
| --- | --- |
| Icahn School of Medicine at Mount Sinai | Paula L Croxson Sean Froudist-Walsh |
| Biotechnology and Biological Sciences Research Council | Rogier B Mars |
| Charles H. Revson Foundation | Paula L Croxson |

The funders had no role in study design, data collection and interpretation, or the decision to submit the work for publication.

### Author contributions

Sean Froudist-Walsh, Resources, Data curation, Software, Formal analysis, Validation, Investigation, Visualization, Methodology, Writing—original draft, Writing—review and editing; Philip GF Browning, James J Young, Kathy L Murphy, Investigation, Methodology, Writing—review and editing; Rogier B Mars, Investigation, Writing—original draft, Writing—review and editing; Lazar Fleysher, Resources, Supervision, Investigation, Methodology; Paula L Croxson, Conceptualization, Resources, Data curation, Formal analysis, Supervision, Funding acquisition, Investigation, Visualization, Methodology, Writing—original draft, Project administration, Writing—review and editing

### Author ORCIDs

Sean Froudist-Walsh http://orcid.org/0000-0003-4070-067X
James J Young https://orcid.org/0000-0001-9349-7519
Rogier B Mars http://orcid.org/0000-0001-6302-8631
Paula L Croxson http://orcid.org/0000-0002-4649-980X

### Ethics

Animal experimentation: This study was conducted ethically, humanely, and in accordance with federal regulations and the guidelines contained in the National Research Council Guide for the Care and Use of Laboratory Animals. All animals were handled according to a protocol approved by the Animal Care and Use Committee (IACUC) at the Icahn School of Medicine at Mount Sinai. The institution also has been fully accredited by the Association for Assessment and Accreditation of Laboratory Animal Care International (AAALAC - #00002) since 1967. All animals procedures are described in detail in the Materials and Methods section of the manuscript.

### Decision letter and Author response

Decision letter https://doi.org/10.7554/eLife.34354.017
Author response https://doi.org/10.7554/eLife.34354.018

# Additional files

## Supplementary files
• Transparent reporting form
DOI: https://doi.org/10.7554/eLife.34354.013

## Data availability

Data are available to download from the INDI PRIMatE Data Exchange (Milham et al., 2018): https://www.nitrc.org/account/login.php?return_to=http://fcon_1000.projects.nitrc.org/indi/PRIMEdownloads.html as the Mount Sinai Philips Achieva 3T dataset. Users will first be prompted to log on to NITRC and will need to register with the 1000 Functional Connectomes Project website on NITRC (http://fcon_1000.projects.nitrc.org/indi/PRIME/mssm1.html) to gain access to the PRIME-DE datasets. The code used for analysis has been made available on GitHub: https://github.com/seanfw/froudist-walsh-et-al-elife-2018 (copy archived at https://github.com/elifesciences-publications/froudist-walsh-et-al-elife-2018).

The following dataset was generated:

| Author(s) | Year | Dataset title | Dataset URL | Database and Identifier |
|---|---|---|---|---|
| Margulies D, Milham M, Schroeder C | 2017 | Mount Sinai Philips Achieva 3T | https://www.nitrc.org/account/login.php?return_to=http://fcon_1000.projects.nitrc.org/indi/PRIMEdownloads.html | INDI PRIMatE Data Exchange, https://www.nitrc.org/frs/downloadlink.php/10419 |

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
