## [Decision Letter]

Thank you for submitting your article "Macro-connectomics and microstructure predict dynamic plasticity patterns in the non-human primate brain" for consideration by *eLife*. Your article has been reviewed by David Van Essen as the Senior Editor, a Reviewing Editor, and three reviewers. The following individuals involved in review of your submission have agreed to reveal their identity: Mikail Rubinov (Reviewer #1); Rosanna Olsen (Reviewer #2); Katherine Duncan (Reviewer #3).

The reviewers have discussed the reviews with one another and the Reviewing Editor has drafted this decision to help you prepare a revised submission.

The reviewers all felt that the dataset is exciting and unique and that the analyses conducted and presented in the paper provide a comprehensive investigation into neural reorganization following a brain injury.

However, they all agreed that there are improvements to be made to the statistical approach and presentation of the results. First, and foremost, it was noted that the analyses do not capitalize on the longitudinal nature of the dataset. Instead, analyses of the data are treated as if they are all independent even though the 3-month period data is included in both dependent measures and differences across animals not properly accounted for. Thus, it would be important to adopt mixed linear models to account for variance across animals (notably, these models can deal with missing data and allow a repeated measures approach).

The reviewers also agreed that the description of the results was very hard to follow. Reviewer 1 pointed out that the authors have taken a 'kitchen-sink' approach without building a logical flow of data presentation. I won't go into detail here but instead am choosing to append the specific reviews in this case because I think they will be very helpful in your revision.

*Reviewer #1:*

The study presents analyses of a unique dataset; the effects in my view are novel and interesting. Having said this, I find the main network analyses of this data to be quite clunky. The authors have taken a "kitchen-sink" approach by applying a large group of analysis tools to their data, without carefully considering which of these tools may describe the primary or most important effects and which may describe redundant secondary correlations or byproducts. More specifically, the study describes reductions and increases in individual connections, changes in within-module and between-module connectivity, changes of node strength, participation coefficient, coreness, rich clubness and modularity. Many of these measures undoubtedly correlate with each other which paradoxically seeks not to clarify but to obfuscate the general picture. I discuss my main concerns below in more detail.

1) It would be most useful to first ascertain if the authors have observed any simple changes in the signal which may drive the downstream effects. For example, I couldn't find if the authors have examined changes in functional connectivity distributions (as well as the mean overall connectivity) before and after lesioning.

2) A complementary analysis could also consider the presence of a specific subnetwork affected after lesions (presumably this subnetwork would focus on the hippocampus). An objective and data-driven way to achieve this would be to employ the Network-Based Statistic (Zalesky, 2010). Together analyses 1 and 2 would allow us to gain a basic picture of changes in the localized pattern of network organization.

3) Much of the authors' analyses (modularity, participation coefficient, within and between module connectivity) relies on an accurate clustering of the network into modules or communities. However, running Louvain community detection with default parameters often results in issues with the resolution limit (Fortunato, 2007), which may underestimate the total number of modules. The authors should consider the extent to which their observed effects are robust to the number of modules chosen in the algorithm. Can they estimate in a principled way if the number of modules changes before and after lesioning? Again, the default Louvain algorithm produces a fairly arbitrary resolution of module partitioning. See e.g. Newman, 2016 and Fortunato, 2016 for details.

4) It is quite confusing to refer to within-module connectivity and strength as local connectivity and to between-module connectivity and participation coefficient as long-range connectivity. These are distinct concepts which are studied separately and do not necessarily coincide (e.g. individual within-module connections can also be long-range). If the authors truly wish to consider local and long-range connectivity, they should define these concepts directly based on connection length criteria (e.g. Sepulcre, 2010). Alternatively, they should describe the effects in more direct terms (namely, within and between module connectivity).

5)Several other studies have considered the relationship between centrality and cytoarchitectural density, I wonder if the authors are familiar with these results – they should be mentioned and discussed in the paper. See for example Beul, 2015/2017 Scholtens and van den Heuvel, 2014/2015 and Rubinov, 2015.

6) Hubs and modules, rich-club and core-periphery all describe essentially the same properties. See Rubinov (2016) for details. As mentioned above, running many redundant analyses seems not to clarify but conversely to obfuscate the primary effects.

*Reviewer #2:*

This was a novel, comprehensive investigation into neural reorganization following a brain injury. The authors should be commended for undertaking this rigorous and ambitious study. Below I will list some of the limitations as well as some clarification questions.

The paper nicely outlines the motivation to study neural reorganization, both acutely and after recovery, due to local brain lesions. While this question is indeed important, the current data only speak to the nature of brain organization due to lesions to a single region of the brain (the hippocampus). While, in my mind, this focus is well-motivated due to the dramatic effect of hippocampal lesions on memory function, I felt there was a bit of a disconnect between how the investigation was framed at the outset and the specific methods used here. A justification in the Introduction about why the hippocampus was the target of the current investigation is warranted, given the broad readership of *eLife*.

Similarly, the authors speculate that neural plasticity following lesions is highly dependent on the cellular makeup of different brain regions. It seems that the cellular makeup of the lesioned region itself could also drastically determine the nature of the neural reorganization following injury. Thus, the conclusions about how the brain is transformed due to local injury are somewhat limited.

The authors speculate that the patterns of plasticity during the chronic stage could relate to recovery of function. It is also stated in the Materials and methods section that behavioural data on a memory task was reported in Browning, 2012 (Note: I could not find this report as the reference was incomplete). A critical next question is whether these neuroplastic changes reflect cognitive recovery. Could the authors refer to existing studies that have examined the recovery of function in either non-human animals or humans to tie the brain changes observed here to cognitive changes following brain injury?

Changes to the extended hippocampal system have been reported in cases of developmental amnesia (Rosenbaum, Gao et al., 2015; Dziecol et al., 2017). These individuals have alterations to the fornix, mammillary bodies, and thalamus, similar to the grey matter changes reported in the current work.

The authors characterize grey matter changes due to injury, but do not comment on which white matter tracts were also affected. Can the authors determine whether the alveus, fimbria, and fornix were damaged by the lesions (or altered post-injury)?

The authors state that "statistically significant grey matter loss was restricted to subcortical areas that were monosynaptically connected to the hippocampus." However, there seems to be more regions affected than those listed in the results listed in the text. For example, in Figure 1C, LGN also seems to be affected.

It is stated that the hippocampus was the fourth best predictor of acute changes in long-range connectivity, which leads me to wonder what were the top 3 regions? Is it surprising that there are regions that predict connectivity to a greater extent than the region that was directly lesioned?

*Reviewer #3:*

Strengths:

This manuscript contains a detailed analysis of an exciting and rare dataset - a longitudinal record of functional connectivity and grey matter volume changes resulting from circumscribed lesions to the hippocampus. The findings from this well-controlled animal study have clear and important implications for both clinical and basic research questions in humans.

Concerns:

1) Defining the chronic phase as changes between 3 months and 12 months can lead to interpretative challenges. Can the authors provide clear criteria for which changes can be interpreted as recovery and which as further disruption? It seems as though both the pre-lesion and 3-month points would be required to constrain the interpretation. Relatedly, including the 3-month estimates in both dependent measures (0-3month and 3month-12mount) means that these measurements are not independent. E.g., is the relationship between hippocampal connectivity and increases in chronic long-range connectivity evidence that pre-lesion hippocampal connectivity predicts greater recovery or that it predicts greater acute disruption (as was separately reported).

2) The longitudinal repeated measurements could be a real strength of this paper, but the analyses do not capitalize on the design. Instead, all measurements are treated as "independent." I would strongly recommend using mixed linear models to account for, what I anticipate to be, significant differences across animals given the variable population and lesion extent. These models can accommodate missing data, addressing the authors' reasons for not accounting for the repeated measures. Alternatively, the authors could report the key findings in each animal to qualitatively document how reliable the observed relationships are across animals. At a minimum, more detail needs to be added to current analyses descriptions. E.g., are the results across animals averaged to create a signal estimates prior to entry into the GLM? I assume this was done based on the graphs, but it should be stated. I would also encourage authors to consider analyses that track changes across time within animals. The finding that chronic changes in GM volume are related to acute changes in connectivity is very interesting but raises other questions about the temporal dependence of plasticity changes.

3) I found the presentation of results hard to follow. For example, the GM volume results were split into two distant sections. Additionally, partial models were presented in various forms before the full models, despite the full model often being required to accurately interpret the relationships. I would recommend organizing the Results section according to the three dependent measures, beginning each section with the full models and only reporting partial models when necessary. It would also be helpful to include a correlation matrix for the independent variables to address the independence of these metrics before entering them into a model. The table at the end of the Results section did help to pull the sections together, but some reorganization could add clarity throughout.

4) There should be some discussion of how the excitotoxic lesions compare to common causes of brain damage in humans and the degree to which these results could be expected to generalize given these differences.

[Editors' note: further revisions were requested prior to acceptance, as described below.]

Thank you for submitting your article "Macro-connectomics and microstructure predict dynamic plasticity patterns in the non-human primate brain" for consideration by *eLife*. Your article has been reviewed by Eve Marder as the Senior Editor, a Reviewing Editor, and three reviewers. The following individuals involved in review of your submission have agreed to reveal their identity: Mikail Rubinov (Reviewer #1); Rosanna Olsen (Reviewer #2); Katherine Duncan (Reviewer #3).

The reviewers have discussed the reviews with one another and the Reviewing Editor has drafted this decision to help you prepare a revised submission.

This revision was very responsive to many of the concerns noted and the reviewers are still supportive of the paper. However, they also agreed that further statistical analyses of existing data are needed to build confidence that the reported effects are not driven simply by the use of different monkeys at different timepoints. This is absolutely critical to address. Furthermore, a few additional discussion points should also be added.

Essential revisions:

In the original round of revisions, it was requested that the authors adopt some way to account for repeated measures or to assure that there was evidence that the reported effects are not simply driven by the fact that different monkeys are used at the different timepoints.

In the response, it was claimed that mixed models could not be constructed for their primary analyses. The logic presented however is not clear. For example, a mixed model could be constructed by predicting each observation (e.g. acute change in network participation for region 1, region 2, and so on; N.regions x N.monkey rows) by the four factors of interest as fixed effects along with a random intercept (and ideally random slopes corresponding to each fixed effect) grouped by monkey. The results from this approach would indicate how reliable the observed relationships are across monkeys and, thus, how generalizable they may be.

It would be also be important to demonstrate in some way that major conclusions are driven by patterns that can be consistently observed across monkeys. Given the large amount of missing data, it's possible that some of the differences between acute and chronic time points merely reflect differences in the monkey populations with data at those times. Even qualitatively replicating the reported patterns in the two monkeys with full datasets could alleviate this concern.

In response to reviewer 1, you now briefly report very strong changes to the overall connectivity across the brain but these results are not well integrated with the other dependent measures. How do the four predictors relate to overall functional connectivity? One specific question that arises is with respect to hubness, the most reliable factor, similarly predicted within module connectivity and network participation; would it be more parsimonious to interpret this relationship as being with overall connectivity?

Finally, it is challenging for the reader to understand the relationship between the continuous hubness predictor used in the first half of the results and the three categories of hubs presented in the second half. Specifically, do non-hubs, provincial hubs, and connector hubs systematically map onto different levels of 'hubness', and does the categorical partitioning of hubness into these levels explain changes in network participation above and beyond what hubness can? This may be a remnant of your original very comprehensive, but dense, reporting. We recommend adopting one well-motivated representation of centrality, unless there is a good reason to do otherwise?

---

## [Author Response]

Reviewer #1:

The study presents analyses of a unique dataset; the effects in my view are novel and interesting. Having said this, I find the main network analyses of this data to be quite clunky. The authors have taken a "kitchen-sink" approach by applying a large group of analysis tools to their data, without carefully considering which of these tools may describe the primary or most important effects and which may describe redundant secondary correlations or byproducts. More specifically, the study describes reductions and increases in individual connections, changes in within-module and between-module connectivity, changes of node strength, participation coefficient, coreness, rich clubness and modularity. Many of these measures undoubtedly correlate with each other which paradoxically seeks not to clarify but to obfuscate the general picture. I discuss my main concerns below in more detail.1) It would be most useful to first ascertain if the authors have observed any simple changes in the signal which may drive the downstream effects. For example, I couldn't find if the authors have examined changes in functional connectivity distributions (as well as the mean overall connectivity) before and after lesioning.

We have visualized the distributions of connectivity and performed t-tests. There is an overall increase in the functional connectivity strength over the acute period, and a decrease over the chronic period (Author response image 1).

**Author response image 1. respfig1:** Connectivity strength distribution at different stages.

We have added the following statement to the manuscript (subsection “Functional and structural measures of plasticity”).

“Across all pairwise connections between brain regions, there was an overall increase in the functional connectivity strength over the acute stage (t = 9.37, p = 1x10^-22^), and a decrease over the chronic stage (t = -16.85, p = 2x10^-62^).

2) A complementary analysis could also consider the presence of a specific subnetwork affected after lesions (presumably this subnetwork would focus on the hippocampus). An objective and data-driven way to achieve this would be to employ the Network-Based Statistic (Zalesky, 2010). Together analyses 1 and 2 would allow us to gain a basic picture of changes in the localized pattern of network organization.

We performed this analysis. NBS did not detect any significant clusters of altered connections over the acute period. A cluster of decreased connections was detected over the chronic period that included many connections between temporal and occipital cortex and connections of the prefrontal cortex. This largely mirrors the findings that we show elsewhere: that rather than changes to a core subnetwork, there are widespread changes across many brain regions following the lesion. Note however that this result depended heavily on an arbitrary choice of threshold. We also note that NBS cannot deal with missing values, so we were unable to perform a linear mixed model analysis on this data. We feel that the inclusion of this analysis in the paper may hinder our efforts to make the paper less overcrowded.

We have provided complementary information regarding the anatomical specificity of the lesion effects in the following ways:

1) Cortical maps of network participation, within-module connectivity and grey matter change in Figure 3, Figure 4 and Figure 5.

2) Scatter plots in Figure 3, Figure 4 and Figure 5 colored according to the connectivity with the hippocampus (with reference to Figure 2D).

3) Highlighted regions in the scatter plots in Figure 3, Figure 4 and Figure 5.

4) Voxelwise grey matter change images in Figure 5D,H.

5) Analysis of effects in specific modules (Figure 6).

6) Force-directed maps, with highlighted regions in Figure 6B.

3) Much of the authors' analyses (modularity, participation coefficient, within and between module connectivity) relies on an accurate clustering of the network into modules or communities. However, running Louvain community detection with default parameters often results in issues with the resolution limit (Fortunato, 2007), which may underestimate the total number of modules. The authors should consider the extent to which their observed effects are robust to the number of modules chosen in the algorithm. Can they estimate in a principled way if the number of modules changes before and after lesioning? Again, the default Louvain algorithm produces a fairly arbitrary resolution of module partitioning. See e.g. Newman, 2016 and Fortunato, 2016 for details.

We re-ran all analyses for a range of lambdas (resolution parameters 0.8-1.4). We chose these bounds as with λ = 0.8 we found only 2-3 modules. With λ = 1.4 we consistently found modules with only one or two nodes.

Figure 3 (participation coefficient). The results (significant predictors and model) are consistent across all values of λ.

Figure 4 (within-module connectivity). Hubness is a significant predictor at λ = 0.8,0.9,1,1.3,1.4, but is not significant at λ = 1.1 or 1.2. This is explicitly mentioned in the manuscript (subsection “Higher pre-lesion hippocampal functional connectivity is associated with a chronic stage rise in within-module functional connectivity”). All other results are consistent across all lambdas.

“Hubness (t = -2.08, p = 0.04) was a significant predictor of a chronic stage decrease in within-module functional connectivity across most (λ = 0.8-1 and 1.3-1.4), but not all (λ = 1.1, p = 0.053, λ = 1.2, p = 0.075) of the repetitions of the analysis with different resolution parameters, and thus may be viewed as a marginal result.”

Figure 5 (grey matter volume). Results are consistent across all lambdas. In principle, this result could also have varied according to the modules as participation coefficient contributes to the estimation of hubness, and depends on the modules used.

Figure 6 (modules). Dispersion of the parieto-occipital module between the 3 and 12 month scan was apparent at all granularities.

For all lambdas there was a significant difference between modules in the change in within-module connectivity, with the parieto-occipital module being the lowest at both the acute and chronic stages.

At all lambdas, the number of estimated modules was higher in the pre-lesion scan (min=3, max = 6) than in the 12-month post-lesion scan (min = 2, max = 5). The 3 month scan was never estimated to have a higher number of modules than the pre-lesion scan, or to have a lower number of modules than the 12 month post-lesion scan, but would either lie between the number of modules for the other two timepoints, or be equal to either of the other two.

Figure 7 (hubs). The reduction in within-module connectivity was consistently greater in hubs than nonhubs at all lambdas at both the acute and chronic periods. The provincial hubs always either lay between nonhubs and connector hubs or level with one or the other, at both the acute and chronic time periods, when λ was varied.

4) It is quite confusing to refer to within-module connectivity and strength as local connectivity and to between-module connectivity and participation coefficient as long-range connectivity. These are distinct concepts which are studied separately and do not necessarily coincide (e.g. individual within-module connections can also be long-range). If the authors truly wish to consider local and long-range connectivity, they should define these concepts directly based on connection length criteria (e.g. Sepulcre, 2010). Alternatively, they should describe the effects in more direct terms (namely, within and between module connectivity).

A good point. We have now clarified these terms within the paper and no longer refer to them as local connectivity and long-range connectivity, using within module connectivity and network participation coefficient instead.

5) Several other studies have considered the relationship between centrality and cytoarchitectural density, I wonder if the authors are familiar with these results – they should be mentioned and discussed in the paper. See for example Beul, 2015/2017 Scholtens and van den Heuvel, 2014/2015 and Rubinov, 2015.

We have now added the following section to the Discussion section:

“Several studies have recently examined the relationships between cell densities and hub properties yet consistent relationships have yet to emerge, perhaps due to the use of different experimental techniques and definitions of hubs across different species (Beul et al., 2015, 2017; van den Heuvel et al., 2015; Rubinov et al., 2015; Scholtens et al., 2014).

6) Hubs and modules, rich-club and core-periphery all describe essentially the same properties. See Rubinov (2016) for details. As mentioned above, running many redundant analyses seems not to clarify but conversely to obfuscate the primary effects.

This is a good point and we agree that it is desirable to limit the number of analyses for clarity. We have now cut the analysis of the Rich club and the core, and amended the text in the Results section:

“We investigated the effect that the hippocampal lesions had on the macroconnectivity structure by examining the changes in individual modules and hubs, which are considered the canonical forms of integration and segregation, and hallmarks of interareal connectomes (Rubinov, 2016).”

Reviewer #2:

This was a novel, comprehensive investigation into neural reorganization following a brain injury. The authors should be commended for undertaking this rigorous and ambitious study. Below I will list some of the limitations as well as some clarification questions.The paper nicely outlines the motivation to study neural reorganization, both acutely and after recovery, due to local brain lesions. While this question is indeed important, the current data only speak to the nature of brain organization due to lesions to a single region of the brain (the hippocampus). While, in my mind, this focus is well-motivated due to the dramatic effect of hippocampal lesions on memory function, I felt there was a bit of a disconnect between how the investigation was framed at the outset and the specific methods used here. A justification in the Introduction about why the hippocampus was the target of the current investigation is warranted, given the broad readership of eLife.

We agree that this is an important point to address. We have added some test to the Introduction to describe our motivation for the hippocampal lesions:

“We set out to investigate whether it is possible to predict plastic changes following a discrete, specific lesion, using a bilateral excitotoxic lesion of the hippocampus. The hippocampus is a key part of the episodic memory circuit, but the impact of lesions restricted to the hippocampus itself is not always large (e.g. Zola-Morgan and Squire, 1986; Malkova and Mishkin, 2003). Because of the widespread nature of the episodic memory circuit (Aggleton and Brown, 1999), we hypothesized that this may be due to functional plasticity in the form of intact brain regions compensating for the damaged area (a process we previously showed to be critically dependent on cholinergic inputs to inferior temporal cortex following hippocampal disconnection; Browning et al., 2010; Croxson et al., 2012).”

Similarly, the authors speculate that neural plasticity following lesions is highly dependent on the cellular makeup of different brain regions. It seems that the cellular makeup of the lesioned region itself could also drastically determine the nature of the neural reorganization following injury. Thus, the conclusions about how the brain is transformed due to local injury are somewhat limited.

This is a very important point, and we have now clarified in the Discussion section that we may observe different results following a lesion to a different region:

“Given our finding that plasticity following a lesion is highly dependent on the cellular composition of different brain regions, it may be the case that the specific cellular composition of the hippocampus may also have played a role in the patterns of plasticity we see here. The unique connectivity of the hippocampus relative to other brain regions may also be an important factor. This suggests that we may see very different patterns of following lesions to other brain regions. Future studies will be needed to determine if our findings can be generalised to lesions to other brain regions or are specific to the hippocampus.”

The authors speculate that the patterns of plasticity during the chronic stage could relate to recovery of function. It is also stated in the Materials and methods section that behavioural data on a memory task was reported in Browning, 2012 (Note: I could not find this report as the reference was incomplete).

This work has now been uploaded as a preprint, and the citation has been updated (Froudist-Walsh et al., 2018).

A critical next question is whether these neuroplastic changes reflect cognitive recovery. Could the authors refer to existing studies that have examined the recovery of function in either non-human animals or humans to tie the brain changes observed here to cognitive changes following brain injury? Changes to the extended hippocampal system have been reported in cases of developmental amnesia (Rosenbaum, Gao et al., 2015; Dziecol et al., 2017). These individuals have alterations to the fornix, mammillary bodies, and thalamus, similar to the grey matter changes reported in the current work.

Thank you for this interesting observation. We have now added the following section to the Discussion section..

“We observed decreases in the volume of the medial septum, amygdala and posterior parahippocampal cortex. These regions and the white matter tracts connecting them to the hippocampus are also affected in human subjects with developmental amnesia (Dzieciol et al., 2017; Olsen et al., 2013) and in people born very preterm (Ball et al., 2011; Caldinelli et al., 2017; Froudist-Walsh et al., 2017; Salvan et al., 2014; Tseng et al., 2017). Although the severe structural abnormalities associated with developmental amnesia lead to seemingly permanent impairments to episodic memory (Vargha-Khadem et al., 2001), milder damage to this circuit may enable plastic changes in cortical functional connectivity to partially compensate for damage to the core episodic memory circuit (Isaacs et al., 2003; Nosarti and Froudist-Walsh, 2016). In the present study, the incomplete damage to subcortical structures such as the mammillary bodies, fornix and connected thalamic subregions in combination with plastic changes to spared areas may be crucial for the preservation or recovery or anterograde memory abilities (Baxter, 2013; Froudist-Walsh et al., 2018; Mitchell et al., 2008). Nonetheless, we acknowledge a limitation of the study is that our design cannot distinguish between compensatory and maladaptive plasticity.”

The authors characterize grey matter changes due to injury, but do not comment on which white matter tracts were also affected. Can the authors determine whether the alveus, fimbria, and fornix were damaged by the lesions (or altered post-injury)?

The deformation-based morphometry analysis is not really designed to assess white matter changes, and the resolution of the data relative to the small size of the monkey brain make it difficult to tell whether these smaller white matter tracts were affected, unfortunately.

The authors state that "statistically significant grey matter loss was restricted to subcortical areas that were monosynaptically connected to the hippocampus." However, there seems to be more regions affected than those listed in the results listed in the text. For example, in Figure 1C, LGN also seems to be affected.

You are right; there are two further areas that were affected that we did not mention. These were the LGN and the extrastriate visual cortex. However, we have now carried out a different analysis of the grey matter findings using a linear mixed model to take advantage of the longitudinal nature of the data (as suggested by reviewer 3). This analysis (which deals with the issue that the 3-month time point was previously included in two separate analyses) changes to an overlapping but different set of areas, which are described fully in the text and figure legend.

It is stated that the hippocampus was the fourth best predictor of acute changes in long-range connectivity, which leads me to wonder what were the top 3 regions? Is it surprising that there are regions that predict connectivity to a greater extent than the region that was directly lesioned?

This is a great question. Note however that we have now edited the manuscript to make it clearer and to minimise the overall number of analyses, according to comments from the other reviewers, and this analysis no longer appears.

Out of interest, the top 3 regions whose connections predicted acute changes were the ventral anterior visual cortex, the parahippocampal cortex (which includes the rhinal cortex) and the ventral temporal cortex. These regions were the three that had the strongest functional connectivity to the hippocampus before the lesion. This suggests that the acute effects of the hippocampal lesion extend beyond direct connections to the hippocampus, and also affects interactions between other brain regions (particularly connections between areas strongly connected to the hippocampus and tertiary regions), and connections of hub regions.

We have also now included a table in the Materials and methods section of the regions we used and their abbreviations (Table 5).

Reviewer #3:

Strengths:This manuscript contains a detailed analysis of an exciting and rare dataset -- a longitudinal record of functional connectivity and grey matter volume changes resulting from circumscribed lesions to the hippocampus. The findings from this well-controlled animal study have clear and important implications for both clinical and basic research questions in humans.

Thank you.

Concerns:1) Defining the chronic phase as changes between 3 months and 12 months can lead to interpretative challenges. Can the authors provide clear criteria for which changes can be interpreted as recovery and which as further disruption? It seems as though both the pre-lesion and 3-month points would be required to constrain the interpretation.

By using the terms acute and chronic, we mean to simply describe the time period following injury. We don’t intend to assign any relationship between these terms and functional recovery or further disruption. Although it is widely thought that the greatest degree of spontaneous recovery following human brain lesions occurs in the first three months, we acknowledge more recent research (including our own) highlighting that recovery can take place months to years after the insult (Berthier et al., 2011). We have added this clarification to the paper (Results section).

“We do not make any claims as to different rates of behavioral recovery during these stages, and acknowledge that cognitive recovery can occur in either stage following a brain insult (Berthier et al., 2011; Lazar and Antoniello, 2008).”

Relatedly, including the 3-month estimates in both dependent measures (0-3month and 3month-12mount) means that these measurements are not independent. E.g., is the relationship between hippocampal connectivity and increases in chronic long-range connectivity evidence that pre-lesion hippocampal connectivity predicts greater recovery or that it predicts greater acute disruption (as was separately reported).

We have now identified the chronic changes that are independent of the acute changes by performing a linear regression with acute changes as the independent variable and chronic changes as the dependent variable. The residuals thus define the chronic changes that are independent of acute changes. This was done separately for the participation coefficient, within-module connectivity, and grey matter volume analyses, and in all three cases the residual chronic changes were used as the dependent variable for analysis of the chronic stage (Results section, subsection “Prediction of plasticity in acute and chronic stages” and Figure 3, Figure 4 and Figure 5).

2) The longitudinal repeated measurements could be a real strength of this paper, but the analyses do not capitalize on the design. Instead, all measurements are treated as "independent." I would strongly recommend using mixed linear models to account for, what I anticipate to be, significant differences across animals given the variable population and lesion extent. These models can accommodate missing data, addressing the authors' reasons for not accounting for the repeated measures. Alternatively, the authors could report the key findings in each animal to qualitatively document how reliable the observed relationships are across animals. At a minimum, more detail needs to be added to current analyses descriptions. E.g., are the results across animals averaged to create a signal estimates prior to entry into the GLM? I assume this was done based on the graphs, but it should be stated. I would also encourage authors to consider analyses that track changes across time within animals. The finding that chronic changes in GM volume are related to acute changes in connectivity is very interesting but raises other questions about the temporal dependence of plasticity changes.

For the voxelwise deformation-based morphometry analysis, we re-ran the analysis, fitting a linear mixed model at each voxel. We have changed the Materials and methods section and Results section to reflect this.

We have added the following to the Materials and methods section:

“Longitudinal changes in grey matter volume were assessed using a linear mixed-effects model, implemented in Matlab with the FreeSurfer function lme_mass_fit_vw for mass-univariate linear mixed model analysis (Bernal-Rusiel et al., 2013).”

We did not fit a linear mixed model for the analyses of Figure 3, Figure 4 and Figure 5. This would require running the linear mixed model separately for each brain area, in order to identify areas where e.g. hippocampal connectivity is significantly associated with changes in connectivity. That is, the dependent variable would be a 14x1 column of changes in connectivity for a particular area (with one entry for each scan), and this model would be repeated across all 78 areas.

This is different from our primary question of interest, which is whether the pattern of changes across cortex could be predicted by hippocampal connectivity, cell density and hubness, where the dependent variable is a 78x1 column, with one entry per brain area. For these analyses, we used the average connectivity matrix across monkeys for each timepoint. We have added the following text to the Materials and methods section.

3) I found the presentation of results hard to follow. For example, the GM volume results were split into two distant sections. Additionally, partial models were presented in various forms before the full models, despite the full model often being required to accurately interpret the relationships. I would recommend organizing the Results section according to the three dependent measures, beginning each section with the full models and only reporting partial models when necessary. It would also be helpful to include a correlation matrix for the independent variables to address the independence of these metrics before entering them into a model. The table at the end of the Results section did help to pull the sections together, but some reorganization could add clarity throughout.

Thank you for his comment, which we think has helped to strengthen and clarify the manuscript. We have now reorganized the Results section based on this suggestion and removed the partial models from the manuscript.

We have also replaced regular linear regression with stepwise regression, which is more appropriate for non-independent predictors. We note that similar results were obtained when principal components regression on the covariance matrix was performed.

We note that the predictors were not independent, and we present the correlation matrix between predictors below:

Pearson’s correlation between predictors(r)Neuron densityNon-neuronal cell denstiyhubnessHippocampal connectivityNeuron density10.530.290.24Non-neuronal cell denstiy1-0.040.29hubness10.29Hippocampal connectivity1

Table A. Pearson’s correlation between predictors

4) There should be some discussion of how the excitotoxic lesions compare to common causes of brain damage in humans and the degree to which these results could be expected to generalize given these differences.

This is a good point. We have added the following text to the Discussion section:

“The processes underlying the mechanism of an excitotoxic lesion are partly overlapping with those involved in human brain injury. The initial phase of an ischemic event, for example, leads to excitotoxic death via activation of glutamate receptors, as in our deliberate NMDA lesion, but this is only one of a cascade of processes (Cramer, 2008). The mechanisms involved in traumatic brain injury are less similar to our excitotoxic lesions, starting with cerebral edema and increased intracranial pressure, followed by a number of other factors of which glutamate excitotoxicity is just one (Kinoshita, 2016). The strength of a specific NMDA-induced lesion, which spares fibers of passage within or adjacent to the area (Kohler and Schwarcz, 1983; Coffey et al., 1988) is that we can study the effect of damage to a specific are on the rest of the brain.”

[Editors' note: further revisions were requested prior to acceptance, as described below.]

[…] In the original round of revisions, it was requested that the authors adopt some way to account for repeated measures or to assure that there was evidence that the reported effects are not simply driven by the fact that different monkeys are used at the different timepoints.In the response, it was claimed that mixed models could not be constructed for their primary analyses. The logic presented however is not clear. For example, a mixed model could be constructed by predicting each observation (e.g. acute change in network participation for region 1, region 2, and so on; N.regions x N.monkey rows) by the four factors of interest as fixed effects along with a random intercept (and ideally random slopes corresponding to each fixed effect) grouped by monkey. The results from this approach would indicate how reliable the observed relationships are across monkeys and, thus, how generalizable they may be.

We agree that linear mixed models are a powerful approach for longitudinal data, particularly when missing data are concerned. One issue with the proposed model here is that we do not have measures of acute or chronic change for each monkey as (as the reviewers have noted) many monkeys do not have all three scans. We appreciate that the reviewers were mindful of this possibility and had provided an alternative suggestion. We were able to use this approach, which we describe below.

It would be also be important to demonstrate in some way that major conclusions are driven by patterns that can be consistently observed across monkeys. Given the large amount of missing data, it's possible that some of the differences between acute and chronic time points merely reflect differences in the monkey populations with data at those times. Even qualitatively replicating the reported patterns in the two monkeys with full datasets could alleviate this concern.

We agree that this is a very important point, and have repeated the major analyses (Figure 3, Figure 4, Figure 5A-G) using only the data from the two monkeys that have full datasets. There were three statistically significant differences from the analyses in larger group, which we describe below, but in all three cases the interpretation of the result remained the same (e.g. p values became larger or smaller, but did not cross the p<0.05 significance threshold).

The following has been added to the Materials and methods section and Results section:

Results section:

“In order to test whether the above results were due to the fact that, in some cases scans of different monkeys were used at different timepoints, we repeated all analyses with the data from two monkeys with a complete set of pre- and post-lesion scans. We tested whether the β values of the four independent variables (neuron and non-neuronal cell densities, hubness and pre-lesion hippocampal connectivity) for each of the above regression analyses significantly differed between the two datasets using non-parametric statistics (see Methods). Of the 24 β-values assessed, the following differences in β-values between datasets were observed. Hippocampal connectivity as a predictor for chronic changes in participation coefficient (p = 0.034), hubness as a predictor of acute changes in within-module connectivity (p = 0.0004) and hubness as a predictor of chronic changes in grey matter volume (p = 0.019). In all three of these cases the significance of the result, and hence the interpretation did not change i.e. hubness remained a significant predictor of acute changes in within module connectivity in the full and two-monkey datasets, and the two other predictors remained nonsignificant.”

Materials and methods section:

“We tested for differences between the β values in the regressions for the full and two-monkey datasets use non-parametric statistics as follows. Under the null hypothesis, there is no difference between the full and two-monkey datasets. Therefore, we should obtain the same β values for our regressions at the acute and chronic stages regardless of whether the dependent variable data (e.g. acute change in participation coefficient) for each region is taken from the full, or two-monkey dataset. We thus created a null distribution as follows. For each of the 78 brain regions, the acute change in participation coefficient was randomly taken from either the full, or two monkey datasets. We then ran the acute stepwise regression, as in the original analysis. The β values for the four independent variables (neuron density, non-neuronal cell density, hubness and hippocampal connectivity) for the original (full data) regression were then subtracted from the β values for regression with the randomly drawn data, and this difference in β values was saved. This process was repeated 10,000 times with different random draws. Finally, to obtain a two-tailed p-value, the difference between the β values for the regression on the two-monkey data from the β-values for the fulldata regression for each independent variable were compared to the null distribution. If the difference between the full- and two-monkey datasets lay outside the middle 95% of the null distribution, then the β-values were said to be different at the p <0.05 (two-tailed) level. This process was repeated for each of the six dependent variables reported in the paper (acute/chronic stage changes in participation coefficient, within-module connectivity and grey matter volume).”

In response to reviewer 1, you now briefly report very strong changes to the overall connectivity across the brain but these results are not well integrated with the other dependent measures. How do the four predictors relate to overall functional connectivity? One specific question that arises is with respect to hubness, the most reliable factor, similarly predicted within module connectivity and network participation; would it be more parsimonious to interpret this relationship as being with overall connectivity?

We understand that the reviewers here are referring to node strength, or the overall strength of connectivity of individual brain regions (nodes) to all other nodes. Furthermore, we have focused on positive node strength, ignoring negative correlations due to interpretative difficulties. We have therefore taken this suggestion into account and amended the Discussion section to provide clarity on these points.

As the reviewers predicted, hubness was associated with a drop in node strength at both the acute and chronic periods. Furthermore, at the chronic stage, greater neuron density was associated with drop in node strength and pre-lesion hippocampal connectivity was associated with a rise in nodestrength (as was previously shown for within-module connectivity, Figure 4). We have now altered the Discussion section to emphasise the point that hubness was associated with an overall drop in functional connectivity. However, we feel that for the other independent variables, looking at node strength instead of within-module connectivity and participation coefficient obscures some interesting relationships. We have amended the following lines in the Introduction and Discussion section, to make this clearer.

“In contrast, hub regions suffered a general loss of functional connectivity during both the acute and chronic stages.”

“The more a region acted as a hub in the pre-lesion network, the greater the reduction in functional connectivity at both the acute and chronic stages.”

“We found that hub regions were more likely to lose functional connectivity with other regions (reflected in a drop in both within-module functional connectivity and network participation) following a lesion.”

**Author response image 2. respfig2:** Showing the acute and chronic changes in overall node strength, and its relationship with the four independent variables.

Finally, it is challenging for the reader to understand the relationship between the continuous hubness predictor used in the first half of the results and the three categories of hubs presented in the second half. Specifically, do non-hubs, provincial hubs, and connector hubs systematically map onto different levels of 'hubness', and does the categorical partitioning of hubness into these levels explain changes in network participation above and beyond what hubness can? This may be a remnant of your original very comprehensive, but dense, reporting. We recommend adopting one well-motivated representation of centrality, unless there is a good reason to do otherwise?

We appreciate that this analysis may add confusion to the overall paper, rather than providing additional information. We have now removed the analyses relating to the three hub categories.